# LOW-SWITCHING PRIMAL-DUAL ALGORITHMS FOR SAFE REINFORCEMENT LEARNING

## ABSTRACT

Safety is a key challenge in reinforcement learning (RL), especially in real-world applications like autonomous driving and healthcare. To address this, Constrained Markov Decision Processes (CMDPs) are commonly used to incorporate safety constraints while optimizing performance. However, current methods often face significant safety violations during exploration or suffer from high regret, which represents the performance loss compared to an optimal policy. We propose a low-switching primal-dual algorithm that balances regret with bounded constraint violations, drawing on techniques from online learning and CMDPs. Our approach minimizes policy changes through low-switching updates and enhances sample efficiency using Bernstein-based bonuses. This leads to tighter theoretical bounds on regret and safety, achieving a state-of-the-art regret of $\widetilde{O}(\sqrt{SAH^5K}/(\tau-c^0))$, where $S$ and $A$ is the number of states and actions, $H$ is the horizon, $K$ is the number of episodes, and $(\tau-c^0)$ reflects the safety margin of a known existing safe policy. Our method also ensures a $\tilde{O}(1)$ constraint violation and removes unnecessary dependencies on state space $S$ and planning horizon $H$ in the reward regret, offering a scalable solution for constrained RL in complex environments.

## 1 INTRODUCTION

Safety is a critical concern in reinforcement learning (RL), especially in real-world applications such as autonomous driving (Wang et al., 2020), healthcare (Vincent et al., 2014), and industrial automation (Machado et al., 2011). Constrained Markov Decision Processes (CMDPs) (Altman, 1999) are widely used to ensure safety by incorporating safe constraints and safe policies into the decision-making process. These frameworks allow for optimizing performance while limiting risky actions in safety-critical environments, such as preventing collisions in autonomous vehicles or ensuring correct treatment in healthcare. However, during the course of training, many RL methods can experience significant safety violations (Ding et al., 2021; Efroni et al., 2020), particularly when exploring new states or actions. This is highly problematic in real-world systems where even temporary unsafe actions can result in accidents, equipment damage, or hazardous conditions. For instance, in autonomous driving (Calò et al., 2020), safety violations during training might lead to collisions or dangerous maneuvers, while in robotics (Müller et al., 2021), such violations could cause physical harm to equipment or workers. As a result, it is crucial to design methods that guarantee bounded safety constraint violations, ensuring that any violations during training remain within acceptable limits, thereby preventing catastrophic outcomes while maintaining safety throughout the learning process.

Given the need to handle bounded safety constraint violations, several methods, such as those developed by Liu et al. (2021); Bura et al. (2022), achieve $\widetilde{O}(\sqrt{K})$ regret, where $K$ represents the number of learning episodes. Regret measures the difference between the cumulative reward of the optimal policy and the learned policy. However, this regret bound relies heavily on the size of the state space $S$ and the planning horizon $H$. The state space $S$ refers to the set of all possible configurations or conditions the system may encounter. A larger $S$, as seen in complex environments like robotic manipulation or self-driving systems, increases the difficulty of learning due to the need for more data to explore the space. The planning horizon $H$ represents the number of time steps over which decisions are evaluated, with longer horizons making policy learning more challenging due to the need to consider long-term effects. Applications with large state spaces and long horizons,

such as financial planning or autonomous vehicles, often suffer from sample inefficiency in existing methods.

This raises a critical question: *Can we design safe reinforcement learning (RL) algorithms that achieve sample efficiency in large-scale state spaces and long horizons while guaranteeing bounded safety constraint violation with arbitrarily high probability?*

In this paper, we affirmatively address the posed question by proposing a low-switching model-based algorithm, **SLIM** (*Safe Low-Switching Primal-Dual Model-Based Algorithm*), designed for the tabular episodic constrained reinforcement learning (RL) problem. Our algorithm operates within a primal-dual, model-based online framework. In each episode, a safe and effective policy is obtained through multiple iterations of primal-dual updates in a constrained model represented in its Lagrangian form. This policy is then used to gather data for updating the estimated transition model. The low-switching technique, central to our approach, ensures a lazy update of the empirical transition model, reducing both computational cost and the need for frequent state-action pair visitations. This efficiency allows us to leverage advanced techniques from Zhang et al. (2024), originally developed for simple MDP settings, to derive a tighter theoretical bound on regret.

Our contributions are summarized as follows:

- Algorithmically, we introduce the low-switching technique to CMDP algorithms for model updates. Through this, we reduce the computational complexity and enable a tighter analysis on both regret and constraint violation.

- Analytically, we prove that our algorithm **SLIM** is the first one in CMDP that have the regret bound $\tilde{O}(\sqrt{SAH^5K}/(\tau - c^0))$, where $S$ and $A$ is the number of states and actions, $H$ is the horizon, $K$ is the number of episodes, and $(\tau - c^0)$ reflects the safety margin of a known existing safe policy, which greatly reduces the regret bound by a factor of $\sqrt{SH}$ compared to the previously known best results (Liu et al., 2021), while at the same time keeping a bounded constraint violation of $\tilde{O}(1)$ in terms of the length of learning process $K$.

**Related Work**

**Constrained Markov Decision Process (CMDP)**   The Constrained Markov Decision Process (CMDP) (Altman, 1999) is a key model for addressing safety concerns in reinforcement learning (RL). Many existing works on CMDPs employ a primal-dual approach to achieve sublinear regret while maintaining bounded constraint violations (Vaswani et al., 2022; Jain et al., 2022; Paternain et al., 2019; Ding et al., 2020a; Wei et al., 2020; Ding et al., 2020b). Another widely-used method is adapting policy gradient algorithms (Achiam et al., 2017; Tessler et al., 2019; Stooke et al., 2020; Tian et al., 2024). Furthermore, Efroni et al. (2020) introduces a more stringent metric for hard constraint violation, where only positive constraint violations are accumulated. Their approach achieves sublinear regret, constraint violations and hard constraint violation. Recently, Ghosh et al. (2024) extended this idea to a linear setting, obtaining similar results. In practical applications, ensuring strict adherence to safety constraints without violations often requires system-specific assumptions. For instance, Wachi & Sui (2020) assumes regularity in the safety functions, while Amani et al. (2021) presumes knowledge of a safe action for each state. Additionally, Liu et al. (2021); Bura et al. (2022) assume the existence of a known safe policy and its true constraint value, achieving improved regret bounds and constraint violations compared to Efroni et al. (2020). Building on the assumption of a known safe policy and its true constraint value, our work proposes a primal-dual low-switching algorithm, leveraging advanced techniques from standard MDPs. This approach not only improves the regret bound but also maintains a constant constraint violation. A comprehensive comparison with other methods is provided in table 1, where the definitions of regret, constraint violation (CV) and hard constraint violation (hard CV) are given in eqs. (2) and (3).

**Regret bound of episodic tabular MDP**   In the standard episodic tabular MDP setting, Auer et al. (2008) provided an upper bound of $O(\sqrt{S^2AKHD^2})$, while Dann & Brunskill (2015) established a lower bound of $O(\sqrt{SAH^3K})$. Later, Osband & Van Roy (2017) introduced a posterior sampling approach for RL, achieving minimax-optimal regret bounds of $O(\sqrt{SAHK})$ under certain conditions. Azar et al. (2017) further achieved a minimax optimal regret, and Jin et al. (2018) developed a

UCB-type Q-learning method, improving the regret to $O(\sqrt{SAH^2K})$ with variance-aware bounds. Recently, Zhang et al. (2024) reduced the burn-in cost using advanced techniques, yet such methods are rarely explored in CMDPs. To our knowledge, this work is the first to incorporate these techniques into CMDPs, aiming to improve performance in constrained settings.

Table 1: Regret and constraint violation comparisons for algorithms on episodic CMDPs

| | Setting | Regret | CV |
|---|---|---|---|
| Efroni et al. (2020) | Tabular CMDP | $\tilde{O}(\sqrt{S^2AH^4K})$ | $\tilde{O}(\sqrt{S^2AH^4K})$ |
| Liu et al. (2021) | Tabular CMDP ($\pi^0, c^0$ known) | $\tilde{O}(\frac{\sqrt{S^3AH^6K}}{\tau-c_0})$ | 0 |
| Liu et al. (2021) | Tabular CMDP ($c^0$ known) | $\tilde{O}(\frac{\sqrt{S^3AH^6K}}{\tau-c^0})$ | O(1) |
| Bura et al. (2022) | Tabular CMDP ($\pi^0, c_0$ known) | $\tilde{O}(\frac{\sqrt{S^2AH^6K}}{\tau-c^0})$ | 0 |
| Ghosh et al. (2024) | Linear CMDP | $\tilde{O}(\sqrt{d^3H^4K})$ | $\tilde{O}(\sqrt{d^3H^4K})$ |
| Ghosh et al. (2024) | Tabular CMDP | $\tilde{O}(\sqrt{S^2AH^4K})$ | $\tilde{O}(\sqrt{S^2AH^4K})$ |
| **SLIM** (Ours) | Tabular CMDP ($\pi^0, c^0$ known) | $\tilde{O}(\frac{\sqrt{SAH^5K}}{\tau-c_0})$ | O(1) |

## 1.1 NOTATION

We introduce a set of notation to be used throughout. Let $e_s$ denote the $s$-th standard basis vector (which has 1 at the $s$-th coordinate and 0 otherwise). For any set $\mathcal{X}, \Delta_{\mathcal{X}}$ represents the set of probability distributions over the set $\mathcal{X}$. For any positive integer $N$, we denote $[N] = \{1, \ldots, N\}$. For any two vectors $x, y \in \mathbb{R}^d$ with the same dimension $d$, we use $xy$ to abbreviate inner product $x^\top y$, e.g. $P_{s,a,h}V^*_{h+1,r}$ is abbr. of $\sum_{s'} P_{s,a,h}(s')V^*_{h+1,r}(s')$. For any integer $S > 0$, any probability vector $p \in \Delta_{[S]}$ and another vector $v = [v_i]_{1 \leq i \leq S}$, we denote by

$$\mathbb{V}(p, v) := \langle p, v^2 \rangle - (\langle p, v \rangle)^2 = \langle p, (v - \langle p, v \rangle 1)^2 \rangle$$

the associated variance, where $v^2 = \left[v_i^2\right]_{1 \leq i \leq S}$ represents element-wise square of $v$. For any two vectors $a = [a_i]_{1 \leq i \leq n}$ and $b = [b_i]_{1 \leq i \leq n}$, the notation $a \geq b$ (resp. $a \leq b$) means $a_i \geq b_i$ (resp. $a_i \leq b_i$) holds simultaneously for all $i$.

## 2 PROBLEM SETUP

We consider a finite-horizon non-stationary constrained Markov Decision Process (MDP) defined by the tuple $M = (\mathcal{S}, \mathcal{A}, H, P, r, c, \tau)$, where $\mathcal{S}$ is the state space, $\mathcal{A}$ is the action space, and $H$ is the horizon length. The unknown transition probability at each time step is denoted by $P_{s,a,h}$, where $P_{s,a,h}(s')$ represents the probability of transitioning to state $s'$ from state $s$ after taking action $a$ at time step $h$. The reward function $r_h : \mathcal{S} \times \mathcal{A} \to [0, 1]$ quantifies the immediate reward the agent receives for taking action $a$ in state $s$ at time step $h$. Similarly, the cost function $c_h : \mathcal{S} \times \mathcal{A} \to [0, 1]$ represents safety violations incurred for the same action. We assume that both the reward and cost functions are known to the agent, though the results can be easily extended to the case where neither function is known. Finally, $\tau \in (0, H]$ is a predefined safety constraint that limits the cumulative cost over the episode.

The agent interacts with the environment over $K$ episodes, each consisting of $H$ steps. At the start of each episode $k$, the agent selects a randomized policy $\pi^k = \{\pi_h^k\}_h$, where at time step $h$ the policy $\pi_h^k : \mathcal{S} \to \Delta_{\mathcal{A}}$ prescribes a distribution over actions conditioned on the current state. The policy is executed with the goal of maximizing the cumulative reward while ensuring that the cumulative cost remains within the safety limit.

The cumulative value at state $s$ and time step $h$, with respect to any function $g : \mathcal{S} \times \mathcal{A} \to \mathbb{R}$, under policy $\pi$, is defined as:

$$V_{h,g}^\pi(s) = \mathbb{E}_{P,\pi} \left[ \sum_{t=h}^H g(S_t, A_t) \middle| S_h = s \right],$$

representing the expected cumulative sum of $g(S_t, A_t)$ from time step $h$ to the end of the episode, given that the process starts in state $s$ at time $h$.

The objective of CMDP is to solve the following constrained optimization problem:

$$\max_\pi V_{1,r}^\pi(s_1) \quad \text{s.t.} \quad V_{1,c}^\pi(s_1) \le \tau, \tag{1}$$

where $V_{1,r}^\pi(s_1)$ is the expected cumulative reward value, and $V_{1,c}^\pi(s_1)$ is the expected cumulative cost value, constrained by the safety threshold $\tau$. The optimal policy that solves eq. (1) is denoted by $\pi^*$ and its corresponding expected reward and cost value are denoted by $V_{1,r}^{\pi^*}(s_1)$ and $V_{1,c}^{\pi^*}(s_1)$.

**Assumption 2.1** (Strictly Safe Policy). *There exists a policy $\pi^0$ such that $V_{1,c}^{\pi^0}(s_1) = c^0 < \tau$, ensuring the policy satisfies strict safety constraints.*

The agent has prior knowledge of a strictly safe policy $\pi^0$ as well as its safety cost value $c^0 = V_{1,c}^{\pi^0}(s_1)$. To understand the agent's objectives, we need to define the regret and constraint violation over $K$ episodes:

$$\text{Regret}(K) := \sum_{k=1}^K \left( V_{1,r}^{\pi^*}(s_1^k) - V_{1,r}^{\pi^k}(s_1^k) \right), \tag{2}$$

$$\text{CV}(K) := \left( \sum_{k=1}^K \left( V_{1,c}^{\pi^k}(s_1^k) - \tau \right) \right)_+. \tag{3}$$

The agent's objective is to minimize regret over $K$ episodes while maintain a low constraint violation.

## 3 METHODOLOGY

We use a model-based approach to address the CMDP problem defined in eq. (1). In contrast to Liu et al. (2021); Bura et al. (2022) where the agent updates an empirical transition model at the end of each episode, we adopt the low-switching technique proposed in Zhang et al. (2024). By using the low-switching technique, we update our empirical transition matrix only when the visitation count of any state-action pair doubles. To be specific, we denote $\bar{N}_h(s, a)$ as the total visitation count of state-action pair $(s, a)$ in time step $h$, $N_h(s, a, s')$ as the count of transitions from $(s, a)$ to $s'$ since the last update, and $N_h(s, a) = \sum_{s'} N_h(s, a, s')$ as the visitation count of $(s, a)$ since the last update. We update an empirical transition matrix $\hat{P}$ whenever $\bar{N}_h(s, a)$ for any $(s, a)$ doubles, such that $\hat{P}_{s,a,h}(s') = \frac{N_h(s,a,s')}{N_h(s,a)}$. Note that we will only use data collected after the last update to calculate $\hat{P}$. With the empirical transition probability matrix $\hat{P}$, we are able to formulate an empirical CMDP.

We will adopt the principle of optimism in the face of uncertainty (OFU) and use a UCB-style bonus for both reward and cost. For any reward function $g$ and policy $\pi$, we define the bonus for a $(s, a, h, k)$ tuple as

$$b_{h,g}^{k,\pi}(s, a) = c_1 \sqrt{\frac{\mathbb{V}(\hat{P}_{s,a,h}, \hat{V}_{h+1,g}^\pi) \log(1/\delta')}{N_h(s,a)}} + c_2 \frac{H \log(1/\delta')}{N_h(s,a)}, \tag{4}$$

where $c_1$ and $c_2$ are constant to be specified later and $\delta' = \delta/(200SAH^2K^2)$ is related to the confidence level $\delta$. For reward, we add this Bernstein-style bonus $b_{h,r}(s, a)$ to $r_h(s, a)$ for each $(s, a)$ to encourage exploration. We denote the optimistically biased reward estimate as $\tilde{r}$, i.e., $\tilde{r}_h(s, a) = r_h(s, a) + b_{h,r}(s, a)$. For safety cost, we subtract a Bernstein-style bonus $b_{h,c}(s, a)$ from $c_h(s, a)$. We denote the optimistically biased cost estimate by $\underline{c}$, i.e., $\underline{c}_h(s, a) = c_h(s, a) - b_{h,c}(s, a)$. By using the optimistically biased cost estimate we will underestimate the cumulative cost. To compensate this and strive to satisfy the safety constraint, we define a pessimistic constraint constant $\tau_k'$ for each episode by subtracting a episode-dependent gap $\Delta_k$ from $\tau$, i.e., $\tau_k' = \tau - \Delta_k$. We will specify the value of $\Delta_k$ later.

We now introduce an empirical CMDP for each episode $K$, defined by $\hat{M}_k = (\mathcal{S}, \mathcal{A}, H, \hat{P}, \tilde{r}, \underline{c}, \tau_k')$, and the corresponding optimization problem:

$$\max_\pi \hat{V}_{1,\tilde{r}}^\pi(s_1) \quad \text{s.t.} \quad \hat{V}_{1,\underline{c}}^\pi(s_1) \le \tau_k' := \tau - \Delta_k. \tag{5}$$

---

**Algorithm 1: SLIM**

---

**Input**  $: \mathcal{S}, \mathcal{A}, H, K, r, c, \pi^0, c^0, c_1 = 460/9, c_2 = 544/9, \eta = \sqrt{1/SAH}, T = SAH,$
$\varepsilon = SAH/K, U = H, \alpha = \sqrt{K}.$

**Initialization:** $\theta \leftarrow (\tau - c^0)/2, \Delta_k \leftarrow 2\sqrt{SAH^3/k}$, for all $(s, a, s', h) \in \mathcal{S} \times \mathcal{A} \times \mathcal{S} \times [H]$,
set $N_h(s, a, s') \leftarrow 0, \bar{N}_h(s, a, s') \leftarrow 0, N_h(s, a) \leftarrow 0$; for all $\pi$, set
$\hat{Q}^\pi_{h,\underline{c}}(s, a) \leftarrow 0, \hat{V}^\pi_{h,\underline{c}}(s) \leftarrow 0, \hat{Q}^\pi_{h,\tilde{r}}(s, a) \leftarrow H, \hat{V}^\pi_{h,\tilde{r}}(s) \leftarrow H.$

---

**1** **for** $k = 1, \cdots, K$ **do**
**2**   $\tau'_k = \tau - \Delta_k$
**3**   **for** $t = 1, \cdots, T$ **do**
**4**     $\hat{\pi}^k_t = \arg\max_\pi \hat{V}^\pi_{1,\tilde{r}}(s^k_1) - \frac{\hat{\lambda}^k_t}{\alpha} \hat{V}^\pi_{1,\underline{c}}(s^k_1)$
**5**     $\hat{\lambda}^k_{t+1} = \mathcal{R}_\Lambda [\hat{\lambda}^k_t - \eta(\tau'_k - \hat{V}^{\hat{\pi}^k_t}_{1,\underline{c}}(s^k_1))]$
**6**   $\bar{\pi}^k = \frac{1}{T} \sum_{t=1}^T \hat{\pi}^k_t$
**7**   **if** $\left| \hat{V}^{\pi^0}_{1,\underline{c}}(s^k_1) - c^0 \right| > \theta$ **then**
**8**     $\pi^k = \pi^0$
**9**   **else**
**10**     $\pi^k = \bar{\pi}^k$
**11**   **for** $h = 1, \cdots, H$ **do**
**12**     Observe $s^k_h$, take action $a^k_h \sim \pi^k_h(\cdot|s^k_h)$, receive $r^k_h, c^k_h$, observe $s^k_{h+1}$
**13**     $(s, a, s') \leftarrow s^k_h, a^k_h, s^k_{h+1}$
**14**     $\bar{N}_h(s, a) \leftarrow \bar{N}_h(s, a) + 1, N_h(s, a, s') \leftarrow N_h(s, a, s') + 1$
**15**     **if** $\bar{N}_h(s, a) \in \{1, 2, 4, \cdots, 2^{\log_2 K}\}$ **then**
**16**       $N_h(s, a) \leftarrow \sum_{\tilde{s}} N_h(s, a, \tilde{s})$
**17**       $\hat{P}_{s,a,h}(\tilde{s}) \leftarrow N_h(s, a, \tilde{s})/N_h(s, a)$
**18**       TRIGGERED $\leftarrow$ TRUE
**19**       $N_h(s, a, \cdot) \leftarrow 0$
**20**   **if** *TRIGGERED* **then**
**21**     TRIGGERED $\leftarrow$ FALSE
**22**     $\hat{V}^\pi_{H+1,g}(s) \leftarrow 0, \forall x \in \mathcal{S}$
**23**     **for** $h = H, H-1, \cdots, 1$ **do**
**24**       **for** $(s, a) \in \mathcal{S} \times \mathcal{A}$ *and any* $\pi$ **do**
**25**         $\hat{Q}^\pi_{h,\tilde{r}}(s, a) = \min\{r_h(s, a) + b^{k,t,\pi}_{h,r}(s, a) + \hat{P}_{s,a,h} \hat{V}^\pi_{h+1,\tilde{r}}, H\}$
**26**         $\hat{V}^\pi_{h,\tilde{r}}(s) = \sum_{a \in \mathcal{A}} \pi(a|s) \hat{Q}^\pi_{h,\tilde{r}}(s, a)$
**27**         $\hat{Q}^\pi_{h,\underline{c}}(s, a) = \max\{c_h(s, a) - b^{k,t,\pi}_{h,c}(s, a) + \hat{P}_{s,a,h} \hat{V}^\pi_{h+1,\underline{c}}, 0\}$
**28**         $\hat{V}^\pi_{h,\underline{c}}(s) = \sum_{a \in \mathcal{A}} \pi(a|s) \hat{Q}^\pi_{h,\underline{c}}(s, a)$

---

To solve the empirical CMDP problem defined in eq. (5), we employ a primal-dual approach. This method transforms the constrained optimization problem into a saddle-point problem, where we aim to maximize the reward while minimizing constraint violations. Let $\lambda \geq 0$ be the dual variable associated with the cost constraint. The Lagrangian for the empirical CMDP is defined as:

$$\mathcal{L}(\pi, \lambda) = \hat{V}^\pi_{1,\tilde{r}}(s_1) - \lambda \left( \hat{V}^\pi_{1,\underline{c}}(s_1) - \tau'_k \right),$$

where $\pi$ is the primal variable representing the policy, and $\lambda$ is the dual variable penalizing the constraint violation. The equivalent saddle-point problem to eq. (5) is:

$$\min_{\lambda \geq 0} \max_\pi \hat{V}^\pi_{1,\tilde{r}}(s_1) - \lambda \left( \hat{V}^\pi_{1,\underline{c}}(s_1) - \tau'_k \right). \tag{6}$$

In this formulation, the policy $\pi$ seeks to maximize the cumulative reward $\hat{V}^\pi_{1,\tilde{r}}(s_1)$, while the dual variable $\lambda$ penalizes any violation of the cost constraint. Denote $(\hat{\pi}^{k,*}, \hat{\lambda}^{k,*})$ as the optimal solutions to the saddle point problem *eq.* (6).

We solve the saddle-point problem eq. (6) iteratively, and for each iteration $t \in [T]$, we alternatively update iterates of the primal variable $\hat{\pi}_t^k$ and the dual variable $\hat{\lambda}_t^k$. The primal update involves solving the maximization problem over $\pi$,

$$\hat{\pi}_t^k = \arg\max_\pi \hat{V}_{1,\tilde{r}}^\pi(s_1) - \frac{\hat{\lambda}_t^k}{\alpha}\left(\hat{V}_{1,\underline{c}}^\pi(s_1) - \tau_k'\right) = \arg\max_\pi \hat{V}_{1,\tilde{r}}^\pi(s_1) - \frac{\hat{\lambda}_t^k}{\alpha}\hat{V}_{1,\underline{c}}^\pi(s_1), \qquad (7)$$

where $\alpha$ is a constant used to control the cumulative error over $T$ iterations in each episode. The dual update is essentially a gradient descent step with a step size $\eta$. For some technical reasons to be explained later in the proof of lemma A.6, we will round the gradient descent result to the nearest element in an $\varepsilon$-net $\Lambda = \{0, \varepsilon, 2\varepsilon, \ldots, U\}$. Putting everything together, we give the dual update as

$$\hat{\lambda}_{t+1}^k = \mathcal{R}_\Lambda\left[\hat{\lambda}_t^k + \eta\left(\hat{V}_{1,\underline{c}}^{\hat{\pi}_t^k}(s_1) - \tau_k'\right)\right], \qquad (8)$$

where $\mathcal{R}_\Lambda(\lambda) = \arg\min_{p \in \Lambda}|p - \lambda|$ is a rounding function.

Finally, We state our algorithm in alg. 1. We execute $T$ iterations of primal and dual updates from line 3 to 5. Since the bonus terms and gap between empirical model $\hat{P}$ and true transition model $P$ will shrink as we collect more data through the learning process, the gap between the estimate value and true value will shrink. If the gap is larger than certain threshold, i.e., $\left|\hat{V}_{1,\underline{c}}^{\pi^0}(s_1^k) - c^0\right| > \theta$, then we conclude that we do not have a sufficiently accurate empirical model and we execute $\pi^0$ to avoid large constraint violation. If instead we have a good estimate on the transition indicated by the bounded gap, then we execute the mixture policy $\bar{\pi}^k$ obtained from the primal-dual updates.

## 4 MAIN RESULTS AND ANALYSIS

We present the regret and constraint violation bounds of our algorithm and proofs in this section, while we leave intermediate lemmas and proofs used to support the main results in the appendix.

### 4.1 REGRET AND CONSTRAINT VIOLATION RESULTS

**Theorem 4.1.** *With probability at least $1 - \delta$, the regret of alg. 1 is*

$$Regret(K) = \tilde{O}(\sqrt{SAH^5K}/(\tau - c^0)).$$

*Proof.* We decompose the regret as:

$$\begin{aligned}
\text{Regret}(K) &= \sum_{k=1}^K V_{1,r}^*(s_1^k) - V_{1,r}^{\pi^k}(s_1^k) \\
&= \sum_{k=1}^{K_1}\left(V_{1,r}^*(s_1^k) - V_{1,r}^{\pi^0}(s_1^k)\right) + \sum_{k=K_1}^K\left(V_{1,r}^*(s_1^k) - V_{1,r}^{\bar{\pi}^k}(s_1^k)\right),
\end{aligned}$$

where $K_1$ is the number of episodes that the agent chooses $\pi^0$. We note that $\pi^*$ is the optimal solution to the original CMDP optimization problem eq. (1), while for each episode $\bar{\pi}^k$ is an approximation solution to the empirical CMDP optimization problem eq. (5). To cope with the gap between the two policies, we introduce a proxy policy $\pi^{\Delta_k,*}$ that is the optimal solution to the following optimization problem

$$\pi^{\Delta_k,*} \in \arg\max_\pi V_{1,r}^\pi(s_1^k), \quad \text{s.t.} \quad V_{1,c}^\pi(s_1^k) \leq \tau_k' = \tau - \Delta_k. \qquad (9)$$

We now further decompose the regret as

$$\text{Regret}(K) \le \sum_{k=1}^{K_1} \left( V_{1,r}^*(s_1^k) - V_{1,r}^{\pi^0}(s_1^k) \right) + \sum_{k=1}^{K} \left( V_{1,r}^*(s_1^k) - V_{1,r}^{\pi^{\Delta_k,*}}(s_1^k) \right)$$

$$+ \sum_{k=1}^{K} \left( V_{1,r}^{\pi^{\Delta_k,*}}(s_1^k) - \hat{V}_{1,\tilde{r}}^{\pi^{\Delta_k,*}}(s_1^k) \right) + \sum_{k=1}^{K} \left( \hat{V}_{1,\tilde{r}}^{\pi^{\Delta_k,*}}(s_1^k) - \hat{V}_{1,\tilde{r}}^{\bar{\pi}^k}(s_1^k) \right)$$

$$+ \sum_{k=1}^{K} \left( \hat{V}_{1,\tilde{r}}^{\bar{\pi}^k}(s_1^k) - V_{1,r}^{\pi^k}(s_1^k) \right).$$

We give the regret bound by bounding each term above. For the first term we have $\sum_{k=1}^{K_1}(V_{1,r}^*(s_1^k) - V_{1,r}(s_1^k)) \le HK_1 = \tilde{O}(S^2AH^4/(\tau - c^0)^2)$, which is a low-order term and only contributes to the burn-in cost as $K$ is large. The second term is the error incurred by replacing the original constraint constant $\tau$ by a more restrictive empirical constraint constant $\tau_k' = \tau - \Delta_k$ for each episode $k$. We bound the second term in lemma A.1 and with the choice of $\Delta_k = \tilde{O}(\sqrt{SAH^3/k})$, we have

$$\sum_{k=1}^{K} \left( V_{1,r}^*(s_1^k) - V_{1,r}^{\pi^{\Delta_k,*}}(s_1^k) \right) \le \tilde{O}\left( \frac{\sqrt{SAH^5K}}{\tau - c^0} \right). \tag{10}$$

By definition of the proxy policy $\pi^{\Delta_k,*}$ in eq. (9), since $\Delta_k$ is a predetermined constant for each episode $k$, we can see that $\pi^{\Delta_k,*}$ is a deterministic policy that is independent of the online learning process. Thus we can apply lemma A.15 and bound

$$\sum_{k=1}^{K} \left( V_{1,r}^{\pi^{\Delta_k,*}}(s_1^k) - \hat{V}_{1,\tilde{r}}^{\pi^{\Delta_k,*}}(s_1^k) \right) \le 0. \tag{11}$$

The fourth term is the optimization error, and it is incurred because $\bar{\pi}^k$ is an approximation solution generated by iterative primal-dual updates. We bound this term by using the primal update rules in lemma A.2 and have

$$\sum_{k=1}^{K} \left( \hat{V}_{1,\tilde{r}}^{\pi^{\Delta_k,*}} - \hat{V}_{1,\tilde{r}}^{\bar{\pi}^k} \right) = \tilde{O}(\sqrt{SAH^3K}). $$

Finally, the last term in the regret decomposition is the model prediction error, consisting of the errors caused by inaccurate empirical models and additional bonus terms. Worth mentioning, this term is essentially the same as the entire regret in Zhang et al. (2024) as the algorithms share the similar exploration bonus and update rules for transition models. We state in lemma A.3 the bound

$$\sum_{k=1}^{K} \left( \hat{V}_{1,\tilde{r}}^{\bar{\pi}^k}(s_1^k) - V_{1,r}^{\pi^k}(s_1^k) \right) = \tilde{O}(\sqrt{SAH^3K}). \tag{12}$$

Finally, putting everything together, we conclude our final result: with probability at least $1 - \delta$,

$$\text{Regret}(K) = O\left( \sqrt{\frac{SAH^5K \log^5(SAHK/\delta)}{\tau - c^0}} \right). \tag{13}$$

$\square$

**Theorem 4.2.** *With probability at least $1 - \delta$, the constraint violation of alg. 1 is*

$$CV(K) = O(1).$$

*Proof.* By definition of constraint violation,

$$
\begin{aligned}
\text{CV}(K) &= \left( \sum_{k=1}^{K} V_{1,c}^{\pi^k}(s_1^k) - \tau \right)_+ \\
&= \left( \sum_{k=1}^{K_1} \left( V_{1,c}^{\pi^0}(s_1^k) - \tau \right) + \sum_{k=K_1}^{K} \left( V_{1,c}^{\bar{\pi}^k}(s_1^k) - \tau \right) \right)_+ \\
&\leq \left( \sum_{k=K_1}^{K} \left( V_{1,c}^{\bar{\pi}^k}(s_1^k) - \tau \right) \right)_+ \\
&= \left( \sum_{k=K_1}^{K} \left( V_{1,c}^{\bar{\pi}^k}(s_1^k) - \hat{V}_{1,\underline{c}}^{\bar{\pi}^k}(s_1^k) \right) + \sum_{k=K_1}^{K} \left( \hat{V}_{1,\underline{c}}^{\bar{\pi}^k}(s_1^k) - \tau_k' \right) + \sum_{k=1}^{K} \left( \tau_k' - \tau \right) \right)_+ ,
\end{aligned}
$$

where the inequality is due to the fact that $V_{1,c}^{\pi^0}(s_1^k) = c^0 < \tau$. We upper bound each of the three terms in the last line.

For the first term, by definition of optimistically biased estimates of rewards and cost, we note that the analysis of bounding $\sum_k V_{1,c}^{\bar{\pi}^k}(s_1^k) - \hat{V}_{1,\underline{c}}^{\bar{\pi}^k}(s_1^k)$ and $\sum_k \hat{V}_{1,\tilde{r}}^{\bar{\pi}^k}(s_1^k) - V_{1,r}^{\bar{\pi}^k}(s_1^k)$ are analogous, and mostly identical. Hence, by lemma A.3, we have

$$
\sum_{k=K_1}^{K} \left( V_{1,c}^{\bar{\pi}^k}(s_1^k) - \hat{V}_{1,\underline{c}}^{\bar{\pi}^k}(s_1^k) \right) \leq \tilde{O}(\sqrt{SAH^3K}), \tag{14}
$$

with probability at least $1 - SAHK\delta'$.

The second term is the optimization error in the primal-dual process. We calculate $\bar{\pi}^k$ as an approximate solution to the empirical optimization problem defined in eq. (5). Thus, it is not necessarily satisfied that $\hat{V}_{1,\underline{c}}^{\bar{\pi}^k}(s_1^k) \leq \tau_k'$. We hence return to the analysis of the primal-dual framework, and adapt techniques used in Jain et al. (2022); Vaswani et al. (2022). By lemmas A.11 to A.14, we have

$$
\left( \hat{V}_{1,\underline{c}}^{\bar{\pi}^k}(s_1^k) - \tau_k' \right) \leq \left( \hat{V}_{1,\underline{c}}^{\bar{\pi}^k}(s_1^k) - \tau_k' \right)_+ \leq \frac{B[(\tau - c^0) - (\Delta_k + \theta)]}{[(\tau - c^0) - (\Delta_k + \theta)]C - H},
$$

where $B = \frac{\varepsilon^2 + 2\varepsilon U + \eta^2 H^2}{2\eta} + \frac{U^2}{2\eta T}$. By choosing $\theta = (\tau - c^0)/2$, $\Delta_k = 2\sqrt{SAH^3/k}$, $\varepsilon = SAH/K$, $\eta = \sqrt{SA/HK}$, $U = H$, and $T = HK/SA$, we have

$$
\left( \hat{V}_{1,\underline{c}}^{\bar{\pi}^k}(s_1^k) - \tau_k' \right) \leq \tilde{O}(\sqrt{SAHK}). \tag{15}
$$

For the third term, we recall the definition of $\tau_k'$, and we have

$$
\sum_{k=1}^{K} (\tau_k' - \tau) = -\sum_{k=1}^{K} \Delta_k.
$$

We set $\Delta_k = 2\sqrt{\frac{SAH^3}{k}}$ so that the sum will cancel out the leading positive terms. $\qquad \square$

## 5 CONCLUSION

In this paper, we proposed **SLIM**, a low-switching primal-dual algorithm for constrained reinforcement learning, designed to balance regret minimization with safety guarantees in large-scale, complex environments. Our algorithm incorporates the low-switching technique and primal-dual approach to better account for safety constraints in order to achieve safe exploration in online learning. By leveraging the low-switching technique, we can also reduce the frequency of policy updates, thereby improving computational efficiency while maintaining bounded safety violations.

We analytically proved that **SLIM** achieves a regret bound of $\tilde{O}\left( \sqrt{SAH^5K}/(\tau - c^0) \right)$, outperforming existing CMDP methods by reducing the dependency on the size of the state space and

the planning horizon in terms of reward regret. Additionally, we demonstrated that **SLIM** ensures a constant constraint violation of $\tilde{O}(1)$ with high probability, providing robust safety guarantees throughout the learning process.

Our contributions establish new state-of-the-art results for constrained reinforcement learning, particularly in environments with large state-action spaces and long planning horizons. Future work will focus on extending **SLIM** to more general settings, such as model-free environments and continuous state-action spaces, while exploring potential real-world applications in safety-critical domains like autonomous driving and healthcare.

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

# A APPENDIX

## A.1 REGRET ANALYSIS

**Lemma A.1.**
$$\sum_{k=1}^{K} \left( V_{1,r}^{*}(s_1^k) - V_{1,r}^{\pi^{\Delta_k,*}}(s_1^k) \right) \leq \frac{H}{\tau - c^0} \sum_{k=1}^{K} \Delta_k.$$

*Proof.* For each episode $k$, we define a deterministic policy $\mathring{\pi}^k = (1 - \frac{\Delta_k}{\tau - c^0})\pi^* + \frac{\Delta_k}{\tau - c^0}\pi^0$, and its value function satisfies

$$V_{1,c}^{\mathring{\pi}^k}(s_1^k) = (1 - \frac{\Delta_k}{\tau - c^0})V_{1,c}^{\pi^*}(s_1^k) + \frac{\Delta_k}{\tau - c^0}V_{1,c}^{\pi^0}(s_1^k) \le (1 - \frac{\Delta_k}{\tau - c^0})\tau + \frac{\Delta_k}{\tau - c^0}c^0 = \tau - \Delta_k.$$

Then,

$$V_{1,r}^*(s_1^k) - V_{1,r}^{\pi^{\Delta_k,*}}(s_1^k)$$

$$\le V_{1,r}^*(s_1^k) - V_{1,r}^{\mathring{\pi}^k}(s_1^k)$$

$$= V_{1,r}^*(s_1^k) - ((1 - \frac{\Delta_k}{\tau - c^0})V_{1,r}^*(s_1^k) + \frac{\Delta_k}{\tau - c^0}V_{1,r}^{\pi^0}(s_1^k))$$

$$= \frac{\Delta_k}{\tau - c^0}(V_{1,r}^*(s_1^k) - V_{1,r}^{\pi^0}(s_1^k))$$

$$\le \frac{H}{\tau - c^0}\Delta_k,$$

where the first inequality is due to the definition of $\pi^{\Delta_k,*}$, i.e., for any policy $\pi$, s.t. $V_{1,c}^{\pi}(s_1^k) \le \tau_k' = \tau - \Delta_k$, $V_{1,r}^{\pi^{\Delta_k,*}}(s_1^k) \ge V_{1,r}^{\pi}(s_1^k)$. Adding over $K$ episodes gives us the result

$$\sum_{k=1}^{K}\left(V_{1,r}^*(s_1^k) - V_{1,r}^{\pi^{\Delta_k,*}}(s_1^k)\right) \le \frac{H}{\tau - c^0}\sum_{k=1}^{K}\Delta_k.$$

$\square$

**Lemma A.2.**

$$\sum_{k=1}^{K}\left(\hat{V}_{1,\tilde{r}}^{\pi^{\Delta_k,*}}(s_1^k) - \hat{V}_{1,\tilde{r}}^{\bar{\pi}^k}(s_1^k)\right) = \tilde{O}(\sqrt{SAH^3K}).$$

*Proof.* For any primal-dual iteration $t \in [T]$,

$$\hat{V}_{1,\tilde{r}}^{\pi^{\Delta_k,*}}(s_1^k) - \frac{\hat{\lambda}_t^k}{\alpha}\hat{V}_{1,\underline{c}}^{\pi^{\Delta_k,*}}(s_1^k) \le \hat{V}_{1,\tilde{r}}^{\hat{\pi}_t^k}(s_1^k) - \frac{\hat{\lambda}_t^k}{\alpha}\hat{V}_{1,\underline{c}}^{\hat{\pi}_t^k}(s_1^k).$$

Taking average over $T$ iterations,

$$\frac{1}{T}\sum_{t=1}^{T}\left(\hat{V}_{1,\tilde{r}}^{\pi^{\Delta_k,*}}(s_1^k) - \frac{\hat{\lambda}_t^k}{\alpha}\hat{V}_{1,\underline{c}}^{\pi^{\Delta_k,*}}(s_1^k)\right) \le \frac{1}{T}\sum_{t=1}^{T}\left(\hat{V}_{1,\tilde{r}}^{\hat{\pi}_t^k}(s_1^k) - \frac{\hat{\lambda}_t^k}{\alpha}\hat{V}_{1,\underline{c}}^{\hat{\pi}_t^k}(s_1^k)\right).$$

Note that the mixture policy $\bar{\pi}^k$ is the average policies of $\hat{\pi}_t^k$, we have

$$\hat{V}_{1,\tilde{r}}^{\pi^{\Delta_k,*}}(s_1^k) - \frac{1}{T}\sum_{t=1}^{T}\frac{\hat{\lambda}_t^k}{\alpha}\hat{V}_{1,\underline{c}}^{\pi^{\Delta_k,*}}(s_1^k) \le \hat{V}_{1,\tilde{r}}^{\bar{\pi}^k}(s_1^k) - \frac{1}{T}\sum_{t=1}^{T}\frac{\hat{\lambda}_t^k}{\alpha}\hat{V}_{1,\underline{c}}^{\hat{\pi}_t^k}(s_1^k).$$

Further, we notice that

$$\hat{V}_{1,\underline{c}}^{\pi^{\Delta_k,*}}(s_1^k) \le V_{1,c}^{\pi^{\Delta_k,*}}(s_1^k) \le \tau - \Delta_k.$$

Thus, for any episode $k$,

$$\hat{V}_{1,\tilde{r}}^{\pi^{\Delta_k,*}}(s_1^k) - \hat{V}_{1,\tilde{r}}^{\bar{\pi}^k}(s_1^k)$$

$$= \left(\hat{V}_{1,\tilde{r}}^{\pi^{\Delta_k,*}}(s_1^k) - \frac{1}{T}\sum_{t=1}^{T}\frac{\hat{\lambda}_t^k}{\alpha}\hat{V}_{1,\underline{c}}^{\pi^{\Delta_k,*}}(s_1^k)\right) - \left(\hat{V}_{1,\tilde{r}}^{\bar{\pi}^k}(s_1^k) - \frac{1}{T}\sum_{t=1}^{T}\frac{\hat{\lambda}_t^k}{\alpha}\hat{V}_{1,\underline{c}}^{\hat{\pi}_t^k}(s_1^k)\right)$$

$$+ \frac{1}{T}\sum_{t=1}^{T}\frac{\hat{\lambda}_t^k}{\alpha}\left(\hat{V}_{1,\underline{c}}^{\pi^{\Delta_k,*}}(s_1^k) - \hat{V}_{1,\underline{c}}^{\hat{\pi}_t^k}(s_1^k)\right)$$

$$\leq \frac{1}{T}\sum_{t=1}^{T}\frac{\hat{\lambda}_t^k}{\alpha}(\hat{V}_{1,\underline{c}}^{\pi^{\Delta_k,*}}(s_1^k) - \hat{V}_{1,\underline{c}}^{\hat{\pi}_t^k}(s_1^k))$$

$$\leq \frac{1}{T}\sum_{t=1}^{T}\frac{\hat{\lambda}_t^k}{\alpha}(\tau - \Delta_k - \hat{V}_{1,\underline{c}}^{\hat{\pi}_t^k}(s_1^k))$$

$$\leq \frac{\varepsilon^2 + 2\varepsilon U + \eta^2 H^2}{2\eta\alpha} + \frac{U^2}{2\eta\alpha T},$$

where we apply lemma A.14 in the last inequality. By choosing $\alpha = \sqrt{K}$, $\varepsilon = SAH/K$, $U = H$, $T = SAH$, and $\eta = \sqrt{1/SAH}$, we have

$$\sum_{k=1}^{K}\left(\hat{V}_{1,\tilde{r}}^{\pi^{\Delta_k,*}}(s_1^k) - \hat{V}_{1,\tilde{r}}^{\bar{\pi}^k}(s_1^k)\right) = \tilde{O}(\sqrt{SAH^3K}).$$

$\square$

**Lemma A.3.**

$$\sum_{k=1}^{K}\left(\hat{V}_{1,\tilde{r}}^{\bar{\pi}^k}(s_1^k) - V_{1,r}^{\pi^k}(s_1^k)\right) = \tilde{O}(\sqrt{SAH^3K}).$$

*Proof.* By definition, we write

$$\hat{V}_{h,\tilde{r}}^{\bar{\pi}^k}(s_h^k) = \sum_{a\in\mathcal{A}}\bar{\pi}^k(a|s_h^k)\hat{Q}_{h,\tilde{r}}^{\bar{\pi}^k}(s_h^k, a)$$

$$= \hat{Q}_{h,\tilde{r}}^{\bar{\pi}^k}(s_h^k, a_h^k) + \left(\sum_{a\in\mathcal{A}}\bar{\pi}^k(a|s_h^k)\hat{Q}_{h,\tilde{r}}^{\bar{\pi}^k}(s_h^k, a) - \hat{Q}_{h,\tilde{r}}^{\bar{\pi}^k}(s_h^k, a_h^k)\right)$$

$$\leq r_h(s_h^k, a_h^k) + b_h^k(s_h^k, a_h^k) + \hat{P}_{s_h^k, a_h^k, h}^k\hat{V}_{h+1,\tilde{r}}^{\bar{\pi}^k} + \zeta_h^k$$

$$\leq r_h(s_h^k, a_h^k) + b_h^k(s_h^k, a_h^k) + (\hat{P}_{s_h^k, a_h^k, h}^k - P_{s,a,h})\hat{V}_{h+1,\tilde{r}}^{\bar{\pi}^k} + (P_{s_h^k, a_h^k, h} - \mathbb{1}_{\{s_{h+1}^k\}})\hat{V}_{h+1,\tilde{r}}^{\bar{\pi}^k}$$

$$+ \hat{V}_{h+1,\tilde{r}}^{\bar{\pi}^k}(s_{h+1}^k) + \zeta_h^k,$$

where

$$\zeta_h^k = \left(\sum_{a\in\mathcal{A}}\bar{\pi}^k(a|s_h^k)\hat{Q}_{h,\tilde{r}}^{\bar{\pi}^k}(s_h^k, a) - \hat{Q}_{h,\tilde{r}}^{\bar{\pi}^k}(s_h^k, a_h^k)\right)$$

is a zero-mean random variable conditional on $\bar{\pi}^k$. Then by summing over $H$ time steps and telescoping, we have

$$\hat{V}_{1,\tilde{r}}^{\bar{\pi}^k}(s_1^k) \leq \sum_{h=1}^{H} r_h(s_h^k, a_h^k) + b_h^k(s_h^k, a_h^k) + (\hat{P}_{s_h^k, a_h^k, h}^k - P_{s,a,h})\hat{V}_{h+1,\tilde{r}}^{\bar{\pi}^k}$$

$$+ (P_{s_h^k, a_h^k, h} - \mathbb{1}_{\{s_{h+1}^k\}})\hat{V}_{h+1,\tilde{r}}^{\bar{\pi}^k} + \sum_{h=1}^{H}\zeta_h^k.$$

The term we want to bound is now decomposed as

$$\sum_{k=1}^{K} \left( \hat{V}_{1,\tilde{r}}^{\bar{\pi}^k}(s_1^k) - V_{1,r}^{\pi^k}(s_1^k) \right) \leq \sum_{k=1}^{K} \sum_{h=1}^{H} b_h^k(s_h^k, a_h^k) + \sum_{k=1}^{K} \sum_{h=1}^{H} (\hat{P}_{s_h^k,a_h^k,h}^k - P_{s,a,h}) \hat{V}_{h+1,\tilde{r}}^{\bar{\pi}^k} + \sum_{k=1}^{K} \sum_{h=1}^{H} \zeta_h^k$$
$$+ \sum_{k=1}^{K} \sum_{h=1}^{H} (P_{s_h^k,a_h^k,h} - \mathbb{1}_{\{s_{h+1}^k\}}) \hat{V}_{h+1,\tilde{r}}^{\bar{\pi}^k} + \sum_{k=1}^{K} \left( \sum_{h=1}^{H} r_h(s_h^k, a_h^k) - V_{1,r}^{\pi^k}(s_1^k) \right).$$

We apply lemmas A.4, A.5 and A.7 to A.9, and conclude that with probability $1 - \delta$,

$$\sum_{k=1}^{K} \left( \hat{V}_{1,\tilde{r}}^{\bar{\pi}^k}(s_1^k) - V_{1,r}^{\pi^k}(s_1^k) \right) = O\left( \sqrt{SAH^3 K \log^5 \frac{SAHK}{\delta}} \right).$$

$\square$

**Lemma A.4.** *With probability at least* $1 - 3SAHK\delta'$,

$$\sum_{k=1}^{K} \sum_{h=1}^{H} b_{h,r}^{k,\bar{\pi}^k}(s_h^k, a_h^k) \leq \tilde{O}(\sqrt{SAH^3 K}).$$

*Proof.* By definition of bonus $b_{h,r}^{k,\bar{\pi}^k}(s_h^k, a_h^k)$, we have

$$\sum_{k=1}^{K} \sum_{h=1}^{H} b_{h,r}^{k,\bar{\pi}^k}(s_h^k, a_h^k) = \frac{460}{9} \sum_{k,h} \sqrt{\frac{\mathbb{V}\left(\hat{P}_{s_h^k,a_h^k,h}^k, \hat{V}_{h+1,\tilde{r}}^{\bar{\pi}^k}\right) \log \frac{1}{\delta'}}{N_h^k(s_h^k, a_h^k)}} + \frac{544}{9} \sum_{k,h} \frac{H \log \frac{1}{\delta'}}{N_h^k(s_h^k, a_h^k)}.$$

Applying the Cauchy-Schwarz inequality and lemma A.16, we obtain

$$\sum_{k=1}^{K} \sum_{h=1}^{H} b_{h,r}^{k,\bar{\pi}^k}(s_h^k, a_h^k) \leq \frac{460}{9} \sqrt{\sum_{k,h} \frac{\log \frac{1}{\delta'}}{N_h^k(s_h^k, a_h^k)}} \sqrt{\sum_{k,h} \mathbb{V}\left(\hat{P}_{s_h^k,a_h^k,h}^k, \hat{V}_{h+1,\tilde{r}}^{\bar{\pi}^k}\right)}$$
$$+ \frac{544 H \log \frac{1}{\delta'}}{9} \sum_{k,h} \frac{1}{N_h^k(s_h^k, a_h^k)}$$
$$\leq \frac{460}{9} \sqrt{2SAH (\log_2 K) \left(\log \frac{1}{\delta'}\right) \sum_{k,h} \mathbb{V}\left(\hat{P}_{s_h^k,a_h^k,h}^k, \hat{V}_{h+1,\tilde{r}}^{\bar{\pi}^k}\right)}$$
$$+ \frac{1088}{9} SAH^2 (\log_2 K) \log \frac{1}{\delta'}.$$

Then by lemma A.10, we have the desired result. $\square$

**Lemma A.5.**

$$\sum_{k=1}^{K} \sum_{h=1}^{H} (\hat{P}_{s_h^k,a_h^k,h}^k - P_{s,a,h}) \hat{V}_{h+1,\tilde{r}}^{\bar{\pi}^k} \leq \tilde{O}(\sqrt{SAH^3 K}).$$

*Proof.* Note that given a total profile $\mathcal{I} \in \mathcal{C}$ and dual variable sequence $(\hat{\lambda}_1^k, \ldots, \hat{\lambda}_T^k)$, $\hat{V}_{h+1,\tilde{r}}^{\bar{\pi}^k}$ is determined by

$$\left\{ \hat{P}_{s,a,h'}^{\left(I_{s,a,h'}^k\right)}, r_{h'}^{\left(I_{s,a,h'}^k\right)}(s,a), c_{h'}^{\left(I_{s,a,h'}^k\right)}(s,a) \right\}_{h < h' \leq H, (s,a,k) \in \mathcal{S} \times \mathcal{A} \times [K]},$$

and $\|\hat{V}_{h+1,\tilde{r}}^{\bar{\pi}^k}\|_\infty \leq H$. Thus we can invoke lemma A.6 and also by lemma A.10, we have

$$\sum_{k=1}^{K} \sum_{h=1}^{H} (\hat{P}_{s_h^k,a_h^k,h}^k - P_{s,a,h}) \hat{V}_{h+1,\tilde{r}}^{\bar{\pi}^k} \leq \tilde{O}(\sqrt{SAH^3 K}).$$

$\square$

**Lemma A.6.** *Let us first specify the types of vectors $\{X_{h,s,a}\}$. For each total profile $\mathcal{I} \in \mathcal{C}$ and each dual variable sequence $(\lambda_1, \ldots, \lambda_T) \in \Lambda^T$, consider any set $\{\mathcal{X}_{h,\mathcal{I}}\}_{1 \leq h \leq H}$ obeying: for each $1 \leq h \leq H$,*

- *$\mathcal{X}_{h+1,\mathcal{I}}$ is given by a deterministic function of $\mathcal{I}$ and*

$$\left\{\widehat{P}_{s,a,h'}^{(I_{s,a,h'}^k)}, r_{h'}^{(I_{s,a,h'}^k)}(s,a), c_{h'}^{(I_{s,a,h'}^k)}(s,a)\right\}_{h < h' \leq H, (s,a,k) \in \mathcal{S} \times \mathcal{A} \times [K]};$$

- *$\|X\|_\infty \leq H$ for each vector $X \in \mathcal{X}_{h,\mathcal{I}}$;*

- *$\mathcal{X}_{h,\mathcal{I}}$ is a set of no more than $K + 1$ non-negative vectors in $\mathbb{R}^S$, and contains the all-zero vector 0.*

*Suppose that $K \geq SAH \log_2 K$, and construct a set $\{\mathcal{X}_{h,\mathcal{I}}\}_{1 \leq h \leq H}$ for each $\mathcal{I} \in \mathcal{C}$ satisfying the above properties. Then with probability at least $1 - \delta'$,*

$$\sum_{s,a,h \in \mathcal{S} \times \mathcal{A} \times [H]} \left\langle \widehat{P}_{s,a,h}^{(l)} - P_{s,a,h}, X_{h+1,s,a} \right\rangle \leq \sum_{s,a,h \in \mathcal{S} \times \mathcal{A} \times [H]} \max\left\{\left\langle \widehat{P}_{s,a,h}^{(l)} - P_{s,a,h}, X_{h+1,s,a} \right\rangle, 0\right\}$$

$$\leq \sqrt{\frac{8}{2^{l-2}} \sum_{s,a,h} \mathbb{V}\left(P_{s,a,h}, X_{h+1,s,a}\right) \left(6SAH \log_2^2 K + T \log \frac{|\Lambda|}{\delta'}\right)}$$

$$+ \frac{4H}{2^{l-2}} \left(6SAH \log_2^2 K + T \log \frac{|\Lambda|}{\delta'}\right)$$

*holds simultaneously for all $\mathcal{I} \in \mathcal{C}$, all dual variable sequences, all $2 \leq l \leq \log_2 K + 1$, and all sequences $\{X_{h,s,a}\}_{(s,a,h) \in \mathcal{S} \times \mathcal{A} \times [H]}$ obeying $X_{h,s,a} \in \mathcal{X}_{h+1,\mathcal{I}}, \forall(s,a,h) \in \mathcal{S} \times \mathcal{A} \times [H]$.*

*Proof.* This proof is mostly adapted from the proof to lemma 6 in Zhang et al. (2024). Let us begin by considering any fixed total profile $\mathcal{I} \in \mathcal{C}$, any fixed dual variable sequence $(\lambda_1, \ldots, \lambda_T)$, any fixed integer $l$ obeying $2 \leq l \leq \log_2 K + 1$, and any given feasible sequence $\{X_{h,s,a}\}_{(s,a,h) \in \mathcal{S} \times \mathcal{A} \times [H]}$. Recall that (i) $\widehat{P}_{s,a,h}^{(l)}$ is computed based on the $l$-th batch of data comprising $2^{l-2}$ independent samples; and (ii) each $X_{h+1,s,a}$ is given by a deterministic function of $\mathcal{I}$ and the empirical models for steps $h' \in [h+1, H]$. Consequently, lemma A.17 tells us that: with probability at least $1 - \delta'$, one has

$$\sum_{s,a,h} \left\langle \widehat{P}_{s,a,h}^{(l)} - P_{s,a,h}, X_{h+1,s,a} \right\rangle$$

$$\leq \sqrt{\frac{8}{2^{l-2}} \sum_{s,a,h} \mathbb{V}\left(P_{s,a,h}, X_{h+1,s,a}\right) \log \frac{3 \log_2(SAHK)}{\delta'}} + \frac{4H}{2^{l-2}} \log \frac{3 \log_2(SAHK)}{\delta'}$$

where we view the left-hand side as a martingale sequence from $h = H$ back to $h = 1$. Moreover, given that each $X_{h,s,a}$ has at most $K + 1$ different choices (since we assume $|\mathcal{X}_{h,\mathcal{I}}| \leq K + 1$), there are no more than $(K+1)^{SAH} \leq (2K)^{SAH}$ possible choices of the feasible sequence $\{X_{h,s,a}\}_{(s,a,h) \in \mathcal{S} \times \mathcal{A} \times [H]}$. In addition, it has been shown in Lemma 5 of Zhang et al. (2024) that there are no more than $(4SAHK)^{2SAH} \log_2 K$ possibilities of the total profile $\mathcal{I}$. There are in total $|\Lambda|^T$ different choices of dual variable sequences. Here we see that in order to invoke a union bound on a finite number of dual variable sequences, it is required that we introduce an $\varepsilon$-net $\Lambda$ for the dual variable $\lambda$s. We note that by choosing $U = H$, and $\varepsilon = SAH/K$, we have $|\Lambda| = K/SA$. Taking the union bound over all these choices and replacing $\delta'$ with $\delta'/\left((4SAHK)^{2SAH} \log_2 K (2K)^{SAH} \log_2 K |\Lambda|^T\right)$, we can demonstrate that with probability at least $1 - \delta'$,

$$\sum_{s,a,h} \left\langle \widehat{P}_{s,a,h}^{(l)} - P_{s,a,h}, X_{h+1,s,a} \right\rangle$$

$$\leq \sqrt{\frac{8}{2^{l-2}} \sum_{s,a,h} \mathbb{V}\left(P_{s,a,h}, X_{h+1,s,a}\right) \left(6SAH \log_2^2 K + T \log \frac{|\Lambda|}{\delta'}\right)} + \frac{4H}{2^{l-2}} \left(6SAH \log_2^2 K + T \log \frac{|\Lambda|}{\delta'}\right)$$

holds simultaneously for all $\mathcal{I} \in \mathcal{C}$, all dual variable sequences, all $2 \leq l \leq \log_2 K + 1$, and all feasible sequences $\{X_{h,s,a}\}_{(s,a,h) \in \mathcal{S} \times \mathcal{A} \times [H]}$. Finally, recalling our assumption $0 \in \mathcal{X}_{h+1,\mathcal{I}}$, we see that for every total profile $\mathcal{I}$ and its associated feasible sequence $\{X_{h,s,a}\}$

$$\sum_{s,a,h} \max\left\{\left\langle \widehat{P}_{s,a,h}^{(l)} - P_{s,a,h}, X_{h+1,s,a}\right\rangle, 0\right\} \in \left\{\sum_{s,a,h} \left\langle \widehat{P}_{s,a,h}^{(l)} - P_{s,a,h}, \widetilde{X}_{h+1,s,a}\right\rangle \mid \widetilde{X}_{h+1,s,a} \in \mathcal{X}_{h+1,\mathcal{I}}, \forall (s,a,h)\right\}$$

holds true. Consequently, the uniform upper bound on the right-hand side continues to be a valid upper bound on $\sum_{s,a,h} \max\left\{\left\langle \widehat{P}_{s,a,h}^{(l)} - P_{s,a,h}, X_{h+1,s,a}\right\rangle, 0\right\}$. This concludes the proof. $\qquad \square$

**Lemma A.7.** *With probability at least* $1 - 4\delta' \log(KH)$,

$$\sum_{k=1}^{K}\sum_{h=1}^{H} \zeta_h^k \leq \tilde{O}(\sqrt{H^3 K}).$$

*Proof.* Note that $\zeta_h^k = \left(\sum_{a \in \mathcal{A}} \bar{\pi}^k(a|s_h^k)\hat{Q}_{h,\tilde{r}}^{\bar{\pi}^k}(s_h^k, a) - \hat{Q}_{h,\tilde{r}}^{\bar{\pi}^k}(s_h^k, a_h^k)\right)$ is a zero-mean random variable conditional on $\bar{\pi}^k$ and is upper bounded by constant $H$. By lemma A.17, we have

$$\sum_{k=1}^{K}\sum_{h=1}^{H} \zeta_h^k \leq 2\sqrt{2}\sqrt{\sum_{k=1}^{K}\sum_{h=1}^{H} \mathsf{Var}(\zeta_h^k)\log\frac{1}{\delta'}} + 3H\log\frac{1}{\delta'}$$

$$\leq 2\sqrt{2KH^3 \log\frac{1}{\delta'}} + 3H\log\frac{1}{\delta'}$$

with probability at least $1 - 4\delta' \log(KH)$. $\qquad \square$

**Lemma A.8.** *With probability at least* $1 - SAH^2K^2\delta'$,

$$\sum_{k=1}^{K}\sum_{h=1}^{H}(P_{s_h^k,a_h^k,h} - \mathbf{1}_{s_{h+1}^k})\hat{V}_{h+1,\tilde{r}}^{\bar{\pi}^k} \leq \tilde{O}(\sqrt{H^2 K}).$$

*Proof.* We note that conditional on state-action pair $(s_h^k, a_h^k)$, the vectors $P_{s_h^k,a_h^k,h}$ and $\mathbf{1}_{s_{h+1}^k}$ are both independent of the value function estimate $\hat{V}_{h+1,\tilde{r}}^{\bar{\pi}^k}$. Also, the vector $\mathbf{1}_{s_{h+1}^k}$ has the mean of $P_{s_h^k,a_h^k,h}$. Hence, $(P_{s_h^k,a_h^k,h} - \mathbf{1}_{s_{h+1}^k})\hat{V}_{h+1,\tilde{r}}^{\bar{\pi}^k}$ is a zero-mean random variable bounded by $H$ from above, and we thus apply lemma A.17 and have

$$\sum_{k=1}^{K}\sum_{h=1}^{H}(P_{s_h^k,a_h^k,h} - \mathbf{1}_{s_{h+1}^k})\hat{V}_{h+1,\tilde{r}}^{\bar{\pi}^k} \leq 2\sqrt{2}\sqrt{\sum_{k=1}^{K}\sum_{h=1}^{H} \mathbb{V}\left(P_{s_h^k,a_h^k,h}, \hat{V}_{h+1,\tilde{r}}^{\bar{\pi}^k}\right)\log\frac{1}{\delta'}} + 3H\log\frac{1}{\delta'}$$

with probability at least $1 - SAH^2K^2\delta'$. By lemma A.10, we obtain our lemma. $\qquad \square$

**Lemma A.9.** *With probability at least* $1 - 4\delta' \log(KH)$,

$$\sum_{k=1}^{K}\left(\sum_{h=1}^{H} r_h(s_h^k, a_h^k) - V_{1,r}^{\bar{\pi}^k}(s_1^k)\right) \leq \tilde{O}(\sqrt{H^2 K}).$$

*Proof.* Note that conditional on $\bar{\pi}^k$, $E_k := \sum_{h=1}^{H} r_h(s_h^k, a_h^k) - V_{1,r}^{\bar{\pi}^k}(s_1^k)$ is a zero-mean random variable upper bounded by constant $H$. By lemma A.17, we have

$$\left|\sum_{k=1}^{K} E_k\right| \leq 2\sqrt{2}\sqrt{\sum_{k=1}^{K} \mathsf{Var}(E_k)\log\frac{1}{\delta'}} + 3H\log\frac{1}{\delta'}$$

$$\leq 2\sqrt{2KH^2 \log\frac{1}{\delta'}} + 3H\log\frac{1}{\delta'},$$

with probability at least $1 - 4\delta' \log(KH)$, where the last inequality holds because $|E_k| \leq H$. $\qquad \square$

**Lemma A.10.** *With probability at least* $1 - 6SAHK\delta'$,

$$\sum_{k=1}^{K}\sum_{h=1}^{H}\mathbb{V}\left(\hat{P}^k_{s^k_h,a^k_h,h},\hat{V}^{\bar{\pi}^k}_{h+1,\tilde{r}}\right) \leq \tilde{O}(H^2K + \sqrt{H^5K} + SAH^3),$$

$$\sum_{k=1}^{K}\sum_{h=1}^{H}\mathbb{V}\left(P^k_{s^k_h,a^k_h,h},\hat{V}^{\bar{\pi}^k}_{h+1,\tilde{r}}\right) \leq \tilde{O}(H^2K + \sqrt{H^5K} + SAH^3).$$

*Proof.* This proof is modified from the proof to lemma 11 in Zhang et al. (2024), and we show here the parts where the proofs differ. First we write by direct calculation

$$\sum_{k=1}^{K}\sum_{h=1}^{H}\mathbb{V}\left(\hat{P}^k_{s^k_h,a^k_h,h},\hat{V}^{\bar{\pi}^k}_{h+1,\tilde{r}}\right) = \sum_{k=1}^{K}\sum_{h=1}^{H}\left(\left\langle \hat{P}^k_{s^k_h,a^k_h,h},(\hat{V}^{\bar{\pi}^k}_{h+1,\tilde{r}})^2\right\rangle - \left\langle \hat{P}^k_{s^k_h,a^k_h,h},\hat{V}^{\bar{\pi}^k}_{h+1,\tilde{r}}\right\rangle^2\right)$$

$$= \sum_{k=1}^{K}\sum_{h=1}^{H}\left\langle \hat{P}^k_{s^k_h,a^k_h,h} - P^k_{s^k_h,a^k_h,h},(\hat{V}^{\bar{\pi}^k}_{h+1,\tilde{r}})^2\right\rangle + \sum_{k=1}^{K}\sum_{h=1}^{H}\left\langle P^k_{s^k_h,a^k_h,h} - \mathbf{1}_{s^k_{h+1}},(\hat{V}^{\bar{\pi}^k}_{h+1,\tilde{r}})^2\right\rangle$$

$$+ \sum_{k=1}^{K}\sum_{h=2}^{H}(\hat{V}^{\bar{\pi}^k}_{h,\tilde{r}}(s^k_h))^2 - \sum_{k=1}^{K}\sum_{h=1}^{H}\left\langle \hat{P}^k_{s^k_h,a^k_h,h},\hat{V}^{\bar{\pi}^k}_{h+1,\tilde{r}}\right\rangle^2$$

$$\leq \sum_{k=1}^{K}\sum_{h=1}^{H}\left\langle \hat{P}^k_{s^k_h,a^k_h,h} - P^k_{s^k_h,a^k_h,h},(\hat{V}^{\bar{\pi}^k}_{h+1,\tilde{r}})^2\right\rangle + \sum_{k=1}^{K}\sum_{h=1}^{H}\left\langle P^k_{s^k_h,a^k_h,h} - \mathbf{1}_{s^k_{h+1}},(\hat{V}^{\bar{\pi}^k}_{h+1,\tilde{r}})^2\right\rangle$$

$$+ \sum_{k=1}^{K}\sum_{h=1}^{H}\left(\hat{V}^{\bar{\pi}^k}_{h,\tilde{r}}(s^k_h) + \left\langle \hat{P}^k_{s^k_h,a^k_h,h},\hat{V}^{\bar{\pi}^k}_{h+1,\tilde{r}}\right\rangle\right)\left(\hat{V}^{\bar{\pi}^k}_{h,\tilde{r}}(s^k_h) - \left\langle \hat{P}^k_{s^k_h,a^k_h,h},\hat{V}^{\bar{\pi}^k}_{h+1,\tilde{r}}\right\rangle\right),$$

and since the value function estimates are bounded by $H$,

$$\leq \sum_{k=1}^{K}\sum_{h=1}^{H}\left\langle \hat{P}^k_{s^k_h,a^k_h,h} - P^k_{s^k_h,a^k_h,h},(\hat{V}^{\bar{\pi}^k}_{h+1,\tilde{r}})^2\right\rangle + \sum_{k=1}^{K}\sum_{h=1}^{H}\left\langle P^k_{s^k_h,a^k_h,h} - \mathbf{1}_{s^k_{h+1}},(\hat{V}^{\bar{\pi}^k}_{h+1,\tilde{r}})^2\right\rangle$$

$$+ 2H\sum_{k=1}^{K}\sum_{h=1}^{H}\max\left\{\hat{V}^{\bar{\pi}^k}_{h,\tilde{r}}(s^k_h) - \langle \hat{P}_{s^k_h,a^k_h,h},\hat{V}^{\bar{\pi}^k}_{h+1,\tilde{r}}\rangle, 0\right\}$$

$$\leq \sum_{k=1}^{K}\sum_{h=1}^{H}\left\langle \hat{P}^k_{s^k_h,a^k_h,h} - P^k_{s^k_h,a^k_h,h},(\hat{V}^{\bar{\pi}^k}_{h+1,\tilde{r}})^2\right\rangle + \sum_{k=1}^{K}\sum_{h=1}^{H}\left\langle P^k_{s^k_h,a^k_h,h} - \mathbf{1}_{s^k_{h+1}},(\hat{V}^{\bar{\pi}^k}_{h+1,\tilde{r}})^2\right\rangle$$

$$+ 2H\sum_{k=1}^{K}\sum_{h=1}^{H}\max\left\{\hat{V}^{\bar{\pi}^k}_{h,\tilde{r}}(s^k_h) - \hat{Q}^{\bar{\pi}^k}_{h,\tilde{r}}(s^k_h,a^k_h) + \hat{Q}^{\bar{\pi}^k}_{h,\tilde{r}}(s^k_h,a^k_h) - \langle \hat{P}_{s^k_h,a^k_h,h},\hat{V}^{\bar{\pi}^k}_{h+1,\tilde{r}}\rangle, 0\right\}.$$

By definition of update rule of $\hat{Q}$ functions, we have

$$\leq \sum_{k=1}^{K}\sum_{h=1}^{H}\left\langle \hat{P}^k_{s^k_h,a^k_h,h} - P^k_{s^k_h,a^k_h,h},(\hat{V}^{\bar{\pi}^k}_{h+1,\tilde{r}})^2\right\rangle + \sum_{k=1}^{K}\sum_{h=1}^{H}\left\langle P^k_{s^k_h,a^k_h,h} - \mathbf{1}_{s^k_{h+1}},(\hat{V}^{\bar{\pi}^k}_{h+1,\tilde{r}})^2\right\rangle$$

$$+ 2H\sum_{k=1}^{K}\sum_{h=1}^{H}r_h(s^k_h,a^k_h) + 2H\sum_{k=1}^{K}\sum_{h=1}^{H}b^{k,\bar{\pi}^k}_{h,r}(s^k_h,a^k_h) + 2H\sum_{k=1}^{K}\sum_{h=1}^{H}\max\{\xi^k_h,0\},$$

where $\xi^k_h := \hat{V}^{\bar{\pi}^k}_{h,\tilde{r}}(s^k_h) - \hat{Q}^{\bar{\pi}^k}_{h,\tilde{r}}(s^k_h,a^k_h) = \sum_{a\in\mathcal{A}}\bar{\pi}^k(a|s^k_h)\hat{Q}^{\bar{\pi}^k}_{h,\tilde{r}}(s^k_h,a) - \hat{Q}^{\bar{\pi}^k}_{h,\tilde{r}}(s^k_h,a^k_h)$ is a zero-mean random variable conditional on $\bar{\pi}^k$ bounded by $H$. By the results of lemma 10 and 11 in Zhang et al. (2024), we finally bound

$$\sum_{k=1}^{K}\sum_{h=1}^{H}\mathbb{V}\left(\hat{P}^k_{s^k_h,a^k_h,h},\hat{V}^{\bar{\pi}^k}_{h+1,\tilde{r}}\right) \leq \tilde{O}(H^2K + \sqrt{H^5K} + SAH^3).$$

Similarly we can show that with probability at least $1 - 3SAHK\delta'$,

$$\sum_{k=1}^{K}\sum_{h=1}^{H}\mathbb{V}\left(P_{s_h^k,a_h^k,h},\hat{V}_{h+1,\tilde{r}}^{\bar{\pi}^k}\right) = \sum_{k=1}^{K}\sum_{h=1}^{H}\left\langle P_{s_h^k,a_h^k,h},(V_{h+1}^k)^2\right\rangle - \sum_{k=1}^{K}\sum_{h=1}^{H}\left(\left\langle P_{s_h^k,a_h^k,h},V_{h+1}^k\right\rangle\right)^2$$

$$= \sum_{k=1}^{K}\sum_{h=1}^{H}\left\langle P_{s_h^k,a_h^k,h} - \mathbf{1}_{s_{h+1}^k},(V_{h+1}^k)^2\right\rangle + \sum_{k=1}^{K}\sum_{h=2}^{H}(V_h^k(s_h^k))^2 - \sum_{k=1}^{K}\sum_{h=1}^{H}\left(\left\langle P_{s_h^k,a_h^k,h},V_{h+1}^k\right\rangle\right)^2,$$

and we invoke the similar argument as above,

$$\le \sum_{k=1}^{K}\sum_{h=1}^{H}\left\langle P_{s_h^k,a_h^k,h} - \mathbf{1}_{s_{h+1}^k},(V_{h+1}^k)^2\right\rangle + 2H\sum_{k=1}^{K}\sum_{h=1}^{H}\max\left\{V_h^k(s_h^k) - \left\langle P_{s_h^k,a_h^k,h},V_{h+1}^k\right\rangle,0\right\}$$

$$\le \sum_{k=1}^{K}\sum_{h=1}^{H}\left\langle P_{s_h^k,a_h^k,h} - \mathbf{1}_{s_{h+1}^k},(V_{h+1}^k)^2\right\rangle + 2H\sum_{k=1}^{K}\sum_{h=1}^{H}\max\left\{V_h^k(s_h^k) - \left\langle \hat{P}_{s_h^k,a_h^k,h},V_{h+1}^k\right\rangle,0\right\}$$

$$+ 2H\sum_{k=1}^{K}\sum_{h=1}^{H}\max\left\{\left\langle \hat{P}_{s_h^k,a_h^k,h}^k - P_{s_h^k,a_h^k,h},V_{h+1}^k\right\rangle,0\right\}$$

$$\le \sum_{k=1}^{K}\sum_{h=1}^{H}\left\langle P_{s_h^k,a_h^k,h} - \mathbf{1}_{s_{h+1}^k},(V_{h+1}^k)^2\right\rangle + 2H\sum_{k=1}^{K}\sum_{h=1}^{H}r_h(s_h^k,a_h^k) + 2H\sum_{k=1}^{K}\sum_{h=1}^{H}b_{h,r}^{k,\bar{\pi}^k}(s_h^k,a_h^k)$$

$$+ 2H\sum_{k=1}^{K}\sum_{h=1}^{H}\max\{\xi_h^k,0\} + 2H\sum_{k=1}^{K}\sum_{h=1}^{H}\max\left\{\left\langle \hat{P}_{s_h^k,a_h^k,h}^k - P_{s_h^k,a_h^k,h},V_{h+1}^k\right\rangle,0\right\}$$

By the results of lemma 10 and 11 in Zhang et al. (2024), we finally bound

$$\sum_{k=1}^{K}\sum_{h=1}^{H}\mathbb{V}\left(P_{s_h^k,a_h^k,h},\hat{V}_{h+1,\tilde{r}}^{\bar{\pi}^k}\right) \le \tilde{O}(H^2K + \sqrt{H^5K} + SAH^3).$$

$\square$

## A.2 PRIMAL-DUAL OPTIMIZATION ANALYSIS

**Lemma A.11.** *If* $\left|\hat{V}_{1,\tilde{c}}^{\pi^0}(s_1^k) - V_{1,c}^{\pi^0}(s_1^k)\right| \le \theta$ *holds, then*

$$\hat{\lambda}^{k,*} \le \frac{\alpha H}{(\tau - c^0) - (\Delta_k + \theta)}.$$

*Proof.* Writing the empirical CMDP in eq. (5) in its Lagrangian form,

$$\hat{V}_{1,\tilde{r}}^{\hat{\pi}^{k,*}}(s_1^k) = \max_\pi \min_{\lambda \ge 0} \hat{V}_{1,\tilde{r}}^\pi(s_1^k) - \frac{\lambda}{\alpha}\left(\hat{V}_{1,\underline{c}}^\pi(s_1^k) - \tau_k'\right)$$

Using the linear programming formulation of CMDPs in terms of the state-occupancy measures $\mu$, we know that both the objective and the constraint are linear functions of $\mu$, and strong duality holds w.r.t. $\mu$. Since $\mu$ and $\pi$ have a one-to-one mapping, we can switch the min and the max, implying,

$$\hat{V}_{1,\tilde{r}}^{\hat{\pi}^{k,*}}(s_1^k) = \min_{\lambda \ge 0} \max_\pi \hat{V}_{1,\tilde{r}}^\pi(s_1^k) - \frac{\lambda}{\alpha}\left(\hat{V}_{1,\underline{c}}^\pi(s_1^k) - \tau_k'\right)$$

Since $\hat{\lambda}^{k,*}$ is the optimal dual variable for the empirical CMDP in eq. (5),

$$\hat{V}_{1,\tilde{r}}^{\hat{\pi}^{k,*}}(s_1^k) = \max_\pi \hat{V}_{1,\tilde{r}}^\pi(s_1^k) + \frac{\hat{\lambda}^{k,*}}{\alpha}\left(\hat{V}_{1,\underline{c}}^\pi(s_1^k) - \tau_k'\right)$$

$$\ge \hat{V}_{1,\tilde{r}}^{\pi^0}(s_1^k) - \frac{\hat{\lambda}^{k,*}}{\alpha}\left(\hat{V}_{1,\underline{c}}^{\pi^0}(s_1^k) - \tau_k'\right)$$

$$= \hat{V}_{1,\tilde{r}}^{\pi^0}(s_1^k) + \frac{\hat{\lambda}^{k,*}}{\alpha}\left((\tau_k' - \tau) + (\tau - V_{1,c}^{\pi^0}(s_1^k)) + (V_{1,c}^{\pi^0}(s_1^k) - \hat{V}_{1,\underline{c}}^{\pi^0}(s_1^k))\right)$$

Under the event where $\left| \hat{V}_{1,\underline{c}}^{\pi^0}(s_1^k) - V_{1,c}^{\pi^0}(s_1^k) \right| \leq \theta$ for $\theta < \tau - c^0 - \Delta_k$, then

$$\geq \hat{V}_{1,\tilde{r}}^{\pi^0}(s_1^k) + \frac{\hat{\lambda}^{k,*}}{\alpha}\left(-\Delta_k + (\tau - c^0) - \beta\right).$$

Hence, we have

$$\hat{\lambda}^{k,*} \leq \frac{\alpha(\hat{V}_{1,\tilde{r}}^{\hat{\pi}^{k,*}}(s_1^k) - \hat{V}_{1,\tilde{r}}^{\pi^0}(s_1^k))}{(\tau - c^0) - (\Delta_k + \beta)} \leq \frac{\alpha H}{(\tau - c^0) - (\Delta_k + \beta)}.$$

$\square$

**Lemma A.12.** *(Lemma B.2 of Jain et al. (2022)).* For any $C > \lambda^*$ and any $\tilde{\pi}$ s.t.

$$\hat{V}_{1,\tilde{r}}^{\hat{\pi}^{k,*}}(s_1^k) - \hat{V}_{1,\tilde{r}}^{\tilde{\pi}} + C\left(\hat{V}_{1,\underline{c}}^{\tilde{\pi}}(s_1^k) - \tau_k'\right)_+ \leq B,$$

*we have*

$$\left(\hat{V}_{1,\underline{c}}^{\tilde{\pi}}(s_1^k) - \tau_k'\right)_+ \leq \frac{\alpha B}{C - \hat{\lambda}^{k,*}}.$$

*Proof.* Define $\nu(\gamma) = \max_\pi\{\hat{V}_{1,\tilde{r}}^{\pi}(s_1^k) \mid \hat{V}_{1,\underline{c}}^{\pi}(s_1^k) \leq \tau_k' - \gamma\}$ and note that by definition, $\nu(0) = \hat{V}_{1,\tilde{r}}^{\hat{\pi}^{k,*}}(s_1^k)$, and that $\nu$ is a decreasing function for its argument. Then, for any policy $\pi$ s.t. $\hat{V}_{1,\underline{c}}^{\pi}(s_1^k) \leq \tau_k' - \gamma$, we have

$$\hat{V}_{1,\tilde{r}}^{\pi}(s_1^k) - \frac{\hat{\lambda}^{k,*}}{\alpha}(\hat{V}_{1,\underline{c}}^{\pi}(s_1^k) - \tau_k') \leq \max_\pi \hat{V}_{1,\tilde{r}}^{\pi}(s_1^k) - \frac{\hat{\lambda}^{k,*}}{\alpha}(\hat{V}_{1,\underline{c}}^{\pi}(s_1^k) - \tau_k')$$

$$= \hat{V}_{1,\tilde{r}}^{\hat{\pi}^{k,*}}(s_1^k) - \frac{\hat{\lambda}^{k,*}}{\alpha}(\hat{V}_{1,\underline{c}}^{\hat{\pi}^{k,*}}(s_1^k) - \tau_k')$$

$$= \hat{V}_{1,\tilde{r}}^{\hat{\pi}^{k,*}}(s_1^k) = \nu(0) \quad \text{(by strong duality)}$$

This further implies

$$\nu(0) - \frac{\hat{\lambda}^{k,*}}{\alpha}\gamma \geq \hat{V}_{1,\tilde{r}}^{\pi}(s_1^k) - \frac{\hat{\lambda}^{k,*}}{\alpha}(\hat{V}_{1,\underline{c}}^{\pi}(s_1^k) - \tau_k') - \frac{\hat{\lambda}^{k,*}}{\alpha}\gamma$$

$$= \hat{V}_{1,\tilde{r}}^{\pi}(s_1^k) - \frac{\hat{\lambda}^{k,*}}{\alpha}(\hat{V}_{1,\underline{c}}^{\pi}(s_1^k) - (\tau_k' - \gamma))$$

Since this holds for any policy $\pi$ s.t. $\hat{V}_{1,\underline{c}}^{\pi}(s_1^k) \leq \tau_k' - \gamma$, we have

$$\nu(0) - \frac{\hat{\lambda}^{k,*}}{\alpha}\gamma \geq \max_\pi\{\hat{V}_{1,\tilde{r}}^{\pi}(s_1^k) \mid \hat{V}_{1,\underline{c}}^{\pi}(s_1^k) \leq \tau_k' - \gamma\} = \nu(\gamma),$$

and thus

$$\frac{\hat{\lambda}^{k,*}}{\alpha}\gamma \leq \nu(0) - \nu(\gamma).$$

Now we choose $\tilde{\gamma} = -(\hat{V}_{1,\underline{c}}^{\tilde{\pi}}(s_1^k) - \tau_k')_+,$

$$\frac{C - \lambda^{k,*}}{\alpha}|\tilde{\gamma}| = \frac{\lambda^{k,*}}{\alpha}\tilde{\gamma} + \frac{C}{\alpha}|\tilde{\gamma}|$$

$$\leq \nu(0) - \nu(\tilde{\gamma}) + \frac{C}{\alpha}|\tilde{\gamma}|$$

$$= \hat{V}_{1,\tilde{r}}^{\hat{\pi}^{k,*}}(s_1^k) - \hat{V}_{1,\tilde{r}}^{\tilde{\pi}}(s_1^k) + \frac{C}{\alpha}|\tilde{\gamma}| + \hat{V}_{1,\tilde{r}}^{\tilde{\pi}}(s_1^k) - \nu(\tilde{\gamma})$$

$$= \hat{V}_{1,\tilde{r}}^{\hat{\pi}^{k,*}}(s_1^k) - \hat{V}_{1,\tilde{r}}^{\tilde{\pi}}(s_1^k) + \frac{C}{\alpha}(\hat{V}_{1,\underline{c}}^{\tilde{\pi}}(s_1^k) - \tau_k')_+ + \hat{V}_{1,\tilde{r}}^{\tilde{\pi}}(s_1^k) - \nu(\tilde{\gamma})$$

$$\leq B + \hat{V}_{1,\tilde{r}}^{\tilde{\pi}}(s_1^k) - \nu(\tilde{\gamma}).$$

Now let us bound $\nu(\tilde{\gamma})$:

$$\nu(\tilde{\gamma}) = \max_{\pi}\{\hat{V}_{1,\tilde{r}}^{\pi}(s_1^k) \mid \hat{V}_{1,\underline{c}}^{\pi}(s_1^k) \leq \tau_k' + (\hat{V}_{1,\underline{c}}^{\tilde{\pi}}(s_1^k) - \tau_k')_+\}$$

$$\geq \max_{\pi}\{\hat{V}_{1,\tilde{r}}^{\pi}(s_1^k) \mid \hat{V}_{1,\underline{c}}^{\pi}(s_1^k) \leq \hat{V}_{1,\underline{c}}^{\tilde{\pi}}(s_1^k)\} \quad \text{(tightening the constraint)}$$

$$\geq \hat{V}_{1,\tilde{r}}^{\tilde{\pi}}(s_1^k).$$

Finally,

$$\frac{C - \hat{\lambda}^{k,*}}{\alpha}|\tilde{\gamma}| \leq B \implies (\hat{V}_{1,\underline{c}}^{\tilde{\pi}}(s_1^k) - \tau_k')_+ \leq \frac{\alpha B}{C - \hat{\lambda}^{k,*}}.$$

$\square$

**Lemma A.13.**

$$\hat{V}_{1,\tilde{r}}^{\hat{\pi}^{k,*}}(s_1^k) - \hat{V}_{1,\tilde{r}}^{\bar{\pi}^k}(s_1^k) + \frac{\lambda}{\alpha}\left(\hat{V}_{1,\underline{c}}^{\bar{\pi}^k}(s_1^k) - \tau_k'\right) \leq \frac{1}{T}\sum_{t=1}^{T}\frac{1}{\alpha}(\hat{\lambda}_t^k - \lambda)\left(\tau_k' - \hat{V}_{1,\underline{c}}^{\hat{\pi}_t^k}(s_1^k)\right).$$

*Proof.* For any episode $k$ and any time step $t$ in the primal-dual iterations, the primal update ensures that for any policy $\pi$,

$$\hat{V}_{1,\tilde{r}}^{\hat{\pi}_t^k}(s_1^k) - \frac{\hat{\lambda}_t^k}{\alpha}(\hat{V}_{1,\underline{c}}^{\hat{\pi}_t^k}(s_1^k) - \tau_k') \geq \hat{V}_{1,\tilde{r}}^{\pi}(s_1^k) - \frac{\hat{\lambda}_t^k}{\alpha}(\hat{V}_{1,\underline{c}}^{\pi}(s_1^k) - \tau_k').$$

Let $\pi$ be $\hat{\pi}^{k,*}$, and rearrange:

$$\hat{V}_{1,\tilde{r}}^{\hat{\pi}^{k,*}}(s_1^k) - \hat{V}_{1,\tilde{r}}^{\hat{\pi}_t^k}(s_1^k) \leq \frac{\hat{\lambda}_t^k}{\alpha}(\hat{V}_{1,\underline{c}}^{\hat{\pi}^{k,*}}(s_1^k) - \hat{V}_{1,\underline{c}}^{\hat{\pi}_t^k}(s_1^k)).$$

Note that $\hat{\pi}^{k,*}$ is the solution to the empirical CMDP in eq. (5), thus $\hat{V}_{1,\underline{c}}^{\hat{\pi}^{k,*}}(s_1^k) \leq \tau_k'$, and we have

$$\hat{V}_{1,\tilde{r}}^{\hat{\pi}^{k,*}}(s_1^k) - \hat{V}_{1,\tilde{r}}^{\hat{\pi}_t^k}(s_1^k) \leq \frac{\hat{\lambda}_t^k}{\alpha}(\tau_k' - \hat{V}_{1,\underline{c}}^{\hat{\pi}_t^k}(s_1^k)).$$

Take average over $T$ iterations,

$$\frac{1}{T}\sum_{t=1}^{T}\left(\hat{V}_{1,\tilde{r}}^{\hat{\pi}^{k,*}}(s_1^k) - \hat{V}_{1,\tilde{r}}^{\hat{\pi}_t^k}(s_1^k)\right) \leq \frac{1}{T}\sum_{t=1}^{T}\frac{\hat{\lambda}_t^k}{\alpha}\left(\tau_k' - \hat{V}_{1,\underline{c}}^{\hat{\pi}_t^k}(s_1^k)\right).$$

To use lemma A.14, we rewrite as

$$\frac{1}{T}\sum_{t=1}^{T}\left(\hat{V}_{1,\tilde{r}}^{\hat{\pi}^{k,*}}(s_1^k) - \hat{V}_{1,\tilde{r}}^{\hat{\pi}_t^k}(s_1^k)\right) + \frac{1}{T}\sum_{t=1}^{T}\frac{\lambda}{\alpha}\left(\hat{V}_{1,\underline{c}}^{\hat{\pi}_t^k}(s_1^k) - \tau_k'\right) \leq \frac{1}{T}\sum_{t=1}^{T}\frac{1}{\alpha}(\hat{\lambda}_t^k - \lambda)\left(\tau_k' - \hat{V}_{1,\underline{c}}^{\hat{\pi}_t^k}(s_1^k)\right).$$

Note that $\hat{V}_{1,\tilde{r}}^{\hat{\pi}^{k,*}}(s_1^k)$ is constant throughout $T$ primal-dual iterations, and $\bar{\pi}^k$ is a mixture policy, then

$$\hat{V}_{1,\tilde{r}}^{\hat{\pi}^{k,*}}(s_1^k) - \hat{V}_{1,\tilde{r}}^{\bar{\pi}^k}(s_1^k) + \frac{\lambda}{\alpha}\left(\hat{V}_{1,\underline{c}}^{\bar{\pi}^k}(s_1^k) - \tau_k'\right) \leq \frac{1}{T}\sum_{t=1}^{T}\frac{1}{\alpha}(\hat{\lambda}_t^k - \lambda)\left(\tau_k' - \hat{V}_{1,\underline{c}}^{\hat{\pi}_t^k}(s_1^k)\right).$$

$\square$

**Lemma A.14.** *For any episode $k$, and primal and dual updates in eqs. (7) and (8),*

$$\frac{1}{T}\sum_{t=1}^{T}\left(\hat{\lambda}_t^k - \lambda\right)\left(\tau_k' - \hat{V}_{1,\underline{c}}^{\hat{\pi}_t^k}(s_1^k)\right) \leq \frac{\varepsilon^2 + 2\varepsilon U + \eta^2 H^2}{2\eta} + \frac{U^2}{2\eta T}.$$

*Proof.* In this proof, for the simplicity of notations, we will only focus on primal-dual iterations in an arbitrary episode $k \in [K]$, and thus we will drop all dependency on $k$ when the context is clear. The dual update is given by

$$\hat{\lambda}_{t+1} = \mathcal{R}_\Lambda[\hat{\lambda}_t - \eta(\tau' - \hat{V}_{1,\underline{c}}^{\hat{\pi}_t}(s_1^k))].$$

Particularly, we denote

$$\hat{\lambda}'_{t+1} = P_{[0,U]}[\hat{\lambda}_t - \eta(\tau' - \hat{V}^{\hat{\pi}_t}_{1,\underline{c}}(s^k_1))].$$

First, we shall look at $|\hat{\lambda}_t - \lambda|$:

$$
\begin{aligned}
|\hat{\lambda}_{t+1} - \lambda| = |\mathcal{R}_\Lambda[\hat{\lambda}'_{t+1}] - \lambda| &= |\mathcal{R}_\Lambda[\hat{\lambda}'_{t+1}] - \hat{\lambda}'_{t+1} + \hat{\lambda}'_{t+1} - \lambda| \\
&\le |\mathcal{R}_\Lambda[\hat{\lambda}'_{t+1}] - \hat{\lambda}'_{t+1}| + |\hat{\lambda}'_{t+1} - \lambda| \\
&\le \varepsilon + |\hat{\lambda}'_{t+1} - \lambda|.
\end{aligned}
$$

Take square on both sides,

$$
\begin{aligned}
|\hat{\lambda}_{t+1} - \lambda|^2 &\le \varepsilon^2 + 2\varepsilon|\hat{\lambda}'_{t+1} - \lambda| + |\hat{\lambda}'_{t+1} - \lambda|^2 \\
&\le \varepsilon^2 + 2\varepsilon U + |\hat{\lambda}'_{t+1} - \lambda|^2 \\
&\le \varepsilon^2 + 2\varepsilon U + |\hat{\lambda}_t - \eta(\tau' - \hat{V}^{\hat{\pi}_t}_{1,\underline{c}}(s^k_1)) - \lambda|^2 \\
&= \varepsilon^2 + 2\varepsilon U + |\hat{\lambda}_t - \lambda|^2 - 2\eta(\tau' - \hat{V}^{\hat{\pi}_t}_{1,\underline{c}}(s^k_1))(\hat{\lambda}_t - \lambda) + \eta^2(\tau' - \hat{V}^{\hat{\pi}_t}_{1,\underline{c}}(s^k_1))^2 \\
&\le \varepsilon^2 + 2\varepsilon U + |\hat{\lambda}_t - \lambda|^2 - 2\eta(\tau' - \hat{V}^{\hat{\pi}_t}_{1,\underline{c}}(s^k_1))(\hat{\lambda}_t - \lambda) + \eta^2 H^2.
\end{aligned}
$$

Now we have

$$(\hat{\lambda}_t - \lambda)(\tau' - \hat{V}^{\hat{\pi}_t}_{1,\underline{c}}(s^k_1)) \le \frac{\varepsilon^2 + 2\varepsilon U + \eta^2 H^2}{2\eta} + \frac{|\hat{\lambda}_t - \lambda|^2 - |\hat{\lambda}_{t+1} - \lambda|^2}{2\eta}.$$

By taking average over $T$ iterations and telescoping, we have

$$
\begin{aligned}
\frac{1}{T}\sum_{t=1}^{T}(\hat{\lambda}_t - \lambda)(\tau' - \hat{V}^{\hat{\pi}_t}_{1,\underline{c}}(s^k_1)) &\le \frac{\varepsilon^2 + 2\varepsilon U + \eta^2 H^2}{2\eta} + \frac{|\lambda_1 - \lambda|^2 - |\lambda_{T+1} - \lambda|^2}{2\eta T} \\
&\le \frac{\varepsilon^2 + 2\varepsilon U + \eta^2 H^2}{2\eta} + \frac{|\lambda_1 - \lambda|^2}{2\eta T} \\
&\le \frac{\varepsilon^2 + 2\varepsilon U + \eta^2 H^2}{2\eta} + \frac{U^2}{2\eta T}.
\end{aligned}
$$

$\square$

### A.3 USEFUL LEMMAS

**Lemma A.15** (Optimism). *With probability at least , for any deterministic policy $\pi$, reward function $g$ and $s \in \mathcal{S}, h \in [H]$, we have*

$$\hat{V}^{\pi}_{h,\tilde{g}}(s) \ge V^{\pi}_{h,g}(s) \ge \hat{V}^{\pi}_{h,\underline{g}}(s).$$

*Proof.* First, we define the following function

$$f(p, v, n) := \langle p, v\rangle + \max\left\{\frac{20}{3}\sqrt{\frac{\mathbb{V}(p,v)\log\frac{1}{\delta'}}{n}}, \frac{400}{9}\frac{H\log\frac{1}{\delta'}}{n}\right\}$$

for any vector $p \in \Delta^S$, any non-negative vector $v \in \mathbb{R}^S$ obeying $\|v\|_\infty \le H$, and any positive integer $n$. We claim that

$$f(p, v, n) \text{ is non-decreasing in each entry of } v. \tag{16}$$

To justify this claim, consider any $1 \le s \le S$, and let us freeze $p, n$ and all but the $s$-th entries of $v$. It then suffices to observe that (i) $f$ is a continuous function, and (ii) except for at most two possible choices of $v(s)$ that obey $\frac{20}{3}\sqrt{\frac{V(p,v)\log\frac{1}{\delta'}}{n}} = \frac{400}{9}\frac{H\log\frac{1}{\delta'}}{n}$, one can use the properties of $p$ and $v$ to

calculate

$$
\frac{\partial f(p, v, n)}{\partial v(s)} = p(s) + \frac{20}{3} \mathbb{1} \left\{ \frac{20}{3} \sqrt{\frac{\mathbb{V}(p, v) \log \frac{1}{\delta'}}{n}} \geq \frac{400}{9} \frac{H \log \frac{1}{\delta'}}{n} \right\} \frac{p(s)(v(s) - \langle p, v \rangle) \sqrt{\log \frac{1}{\delta'}}}{\sqrt{n \mathbb{V}(p, v)}}
$$

$$
= p(s) + \mathbb{1} \left\{ \sqrt{n \mathbb{V}(p, v) \log \frac{1}{\delta'}} \geq \frac{20}{3} H \log \frac{1}{\delta'} \right\} \frac{\frac{20}{3} H \log \frac{1}{\delta'}}{\sqrt{n \mathbb{V}(p, v) \log \frac{1}{\delta'}}} \cdot \frac{p(s)(v(s) - \langle p, v \rangle)}{H}
$$

$$
\geq \min \left\{ p(s) + p(s) \frac{(v(s) - \langle p, v \rangle)}{H}, p(s) \right\}
$$

$$
\geq p(s) \min \left\{ \frac{H + v(s) - \langle p, v \rangle}{H}, 1 \right\} \geq 0,
$$

thus establishing the claim. We now proceed to the proof of lemma A.15. Consider any $(h, k, s, a)$, and we divide into two cases.

Case 1: $N_h^k(s, a) \leq 2$. In this case, the following trivial bounds arise directly from the value function initiation:

$$
\hat{Q}_{h,\tilde{g}}^\pi(s, a) = H \geq Q_{h,g}^\pi(s, a) \geq 0 = \hat{Q}_{h,\underline{g}}^\pi(s, a),
$$

$$
\hat{V}_{h,\tilde{g}}^\pi(s) = H \geq V_{h,g}^\pi(s) \geq 0 = \hat{V}_{h,\underline{g}}^\pi(s).
$$

Case 2: $N_h^k(s, a) > 2$. Suppose now that $\hat{Q}_{h+1,\tilde{g}}^\pi \geq Q_{h+1,g}^\pi \geq \hat{Q}_{h+1,\underline{g}}^\pi$, which also implies that $\hat{V}_{h+1,\tilde{g}}^\pi \geq V_{h+1,g}^\pi \geq \hat{V}_{h+1,\underline{g}}^\pi$. If $\hat{Q}_{h,\tilde{g}}^\pi(s, a) = H$, then $\hat{Q}_{h,\tilde{g}}^\pi(s, a) \geq Q_{h,g}^\pi(s, a)$ holds trivially, and hence it suffices to look at the case with $\hat{Q}_{h,\tilde{g}}^\pi(s, a) < H$. According to the update rule, it holds that

$$
\hat{Q}_{h,\tilde{g}}^\pi(s, a)
$$

$$
= g_h(s, a) + \left\langle \widehat{P}_{s,a,h}^k, \hat{V}_{h+1,\tilde{g}}^\pi \right\rangle + c_1 \sqrt{\frac{\mathbb{V}\left(\widehat{P}_{s,a,h}^k, \hat{V}_{h+1,\tilde{g}}^\pi\right) \log \frac{1}{\delta'}}{N_h^k(s, a)}} + c_2 \frac{H \log \frac{1}{\delta'}}{N_h^k(s, a)}
$$
(17)

$$
\geq g_h(s, a) + \frac{48 H \log \frac{1}{\delta'}}{3 N_h^k(s, a)} + f\left(\widehat{P}_{s,a,h}^k, \hat{V}_{h+1,\tilde{g}}^\pi, N_h^k(s, a)\right)
$$

$$
\geq g_h(s, a) + \frac{48 H \log \frac{1}{\delta'}}{3 N_h^k(s, a)} + f\left(\widehat{P}_{s,a,h}^k, V_{h+1,g}^\pi, N_h^k(s, a)\right)
$$

for any $(s, a)$, where the last inequality results from the claim (16) and the hypothesis $\hat{V}_{h+1,\tilde{g}}^\pi \geq V_{h+1,g}^\pi$. Moreover, applying Lemma 19, we have

$$
\mathbb{P}\left\{ \left| \left\langle \widehat{P}_{s,a,h}^k - P_{s,a,h}, V_{h+1,g}^\pi \right\rangle \right| > 2 \sqrt{\frac{\mathbb{V}\left(\widehat{P}_{s,a,h}^k, V_{h+1,g}^\pi\right) \log \frac{1}{\delta'}}{N_h^k(s, a)}} + \frac{14 H \log \frac{1}{\delta'}}{3 N_h^k(s, a)} \right\}
$$

$$
\leq \mathbb{P}\left\{ \left| \left\langle \widehat{P}_{s,a,h}^k - P_{s,a,h}, V_{h+1,g}^\pi \right\rangle \right| > \sqrt{\frac{2 \mathbb{V}\left(\widehat{P}_{s,a,h}^k, V_{h+1,g}^\pi\right) \log \frac{1}{\delta'}}{N_h^k(s, a) - 1}} + \frac{7 H \log \frac{1}{\delta'}}{3 N_h^k(s, a) - 1} \right\} \leq 2\delta'.
$$

This implies that with probability at least $1 - 2\delta'$,

$$
f\left(\widehat{P}_{s,a,h}^k, V_{h+1,g}^\pi, N_h^k(s, a)\right) = \left\langle P_{s,a,h}, V_{h+1,g}^\pi \right\rangle + \left\langle \widehat{P}_{s,a,h}^k - P_{s,a,h}, V_{h+1,g}^\pi \right\rangle
$$

$$
+ \max \left\{ \frac{20}{3} \sqrt{\frac{\mathbb{V}\left(\widehat{P}_{s,a,h}^k, V_{h+1,g}^\pi\right) \log \frac{1}{\delta'}}{N_h^k(s, a)}}, \frac{400}{9} \frac{H \log \frac{1}{\delta'}}{N_h^k(s, a)} \right\}
$$

$$
\geq \left\langle P_{s,a,h}, V_{h+1,g}^\pi \right\rangle.
$$

Substitution into eq. (17) gives: with probability at least $1 - 2\delta'$,

$$\hat{Q}^{\pi}_{h,\tilde{g}}(s,a) \geq g_h(s,a) + \langle P_{s,a,h}, V^{\pi}_{h+1,g} \rangle = Q^{\pi}_{h,g}(s,a).$$

The proof for $Q^{\pi}_{h,g} \geq \hat{Q}^{\pi}_{h,\underline{g}}$ is analogous and we leave out here. $\square$

**Lemma A.16.** *Recall the definition of $N^k_h(s^k_h, a^k_h)$ in alg. 1. It holds that:*

$$\sum_{k=1}^{K}\sum_{h=1}^{H} \frac{1}{\max\{N^k_h(s^k_h, a^k_h), 1\}} \leq 2SAH \log_2 K.$$

*Proof.* In view of the doubling batch update rule, it is easily seen that: for any given $(s,a,h)$,

$$\sum_{k=1}^{K} \frac{1}{\max\{N^k_h(s^k_h, a^k_h), 1\}} \mathbb{1}\left\{(s,a) = (s^k_h, a^k_h)\right\} \leq 2\log_2 K,$$

since each $(s,a,h)$ is associated with at most $\log_2 K$ epochs. Summing over $(s,a,h)$ completes the proof. $\square$

**Lemma A.17** (Freedman's inequality). *Let $(M_n)_{n\geq 0}$ be a martingale such that $M_0 = 0$ and $|M_n - M_{n-1}| \leq c \ (\forall n \geq 1)$ hold for some quantity $c > 0$. Define*

$$Var_n := \sum_{k=1}^{n} \mathbb{E}\left[(M_k - M_{k-1})^2 \big| \mathcal{F}_{k-1}\right]$$

*for every $n \geq 0$, where $\mathcal{F}_k$ is the $\sigma$-algebra generated by $(M_1, \ldots, M_k)$. Then for any integer $n \geq 1$ and any $\epsilon, \delta > 0$, one has*

$$\mathbb{P}\left[|M_n| \geq 2\sqrt{2}\sqrt{Var_n \log\frac{1}{\delta}} + 2\sqrt{\epsilon \log\frac{1}{\delta}} + 2c\log\frac{1}{\delta}\right] \leq 2\left(\log_2\left(\frac{nc^2}{\epsilon}\right) + 1\right)\delta.$$

