# OpenReview forum: "Low-Switching Primal-Dual Algorithms for Safe Reinforcement Learning"
_ICLR.cc/2025/Conference — Submitted to ICLR 2025_

### Official Review · Reviewer_MZm7 · 2024-10-27

**Soundness:** 2
**Presentation:** 2
**Contribution:** 2
**Rating:** 3
**Confidence:** 4

**Summary:**

The authors propose a primal-dual algorithm for online tabular episodic CMDPs with unknown transitions and deterministic known rewards/costs. This algorithm requires in input a strictly safe policy and guarantees sublinear regret and constant constraints violations.

**Strengths:**

The main strength of the work is that the algorithm achieves a better dependence on the state space and the horizon  in the regret bound compared with state-of-the-art algorithm for online CMDPs.

**Weaknesses:**

To me, this work presents several weaknesses, which are highlighted in the following:

1. First, the paper focuses on a specific setting only, that is, online learning in CMDPs where a strictly feasible solution is known to the learner. In real-world scenarios, it is, in general, not the case that a known safe policy is known, nor the associated cost. Thus, papers in this field usually provide additional (and more general) analysis for the case a strictly safe policy is not known (see [Liu et al., 2021] and [Stradi et al. 2024]).

2. While it is interesting that the algorithm achieves a better dependence on state and horizon in the regret bound, it happens at the cost of paying some violations. Furthermore, notice that the violations definition proposed in the paper is weak, since it allows for cancellations during the episodes. To me, assuming the knowledge of a strictly feasible solution should be enough to guarantee the safety property at each episode.

3. The algorithmic ideas seem not particularly novel. The primal-dual setting employed in the work is well-studied in literature, and the same reasoning holds for the optimistic rewards/costs functions (e.g., [Efroni et al. 2020]). Bernstein-style bonuses are generally used for transitions estimations in online MDPs [Jin et al. 2020].

To summarize, the $\sqrt{SH}$ improvement in the regret bound appears to me not enough given the aforementioned limitations. Overall, since this is mainly a theory paper, I believe the contribution is too poor to reach the acceptance threshold for a top conference such as ICLR.

[Liu et al., 2021] "Learning policies with zero or bounded constraint violation for constrained mdps."

[Stradi et al. 2024] "Learning Adversarial MDPs with Stochastic Hard Constraints"

[Efroni et al. 2020] "Exploration-exploitation in constrained mdps"

[Jin et al. 2020] "Learning Adversarial Markov Decision Processes with Bandit Feedback and Unknown Transition"

**Questions:**

See Weaknesses section.

---

### Official Review · Reviewer_GefV · 2024-10-27

**Soundness:** 3
**Presentation:** 3
**Contribution:** 2
**Rating:** 6
**Confidence:** 4

**Summary:**

This paper studies the safe RL problem by formulating it as a constrained MDP.  Given a known base policy $\pi^0$ and its value $c^0$, the authors provide a low-switching no-regret CMDP algorithm and achieve a tighter regret upper bound, with only a constant constraint violation. In particular, they achieve a tighter regret on the dependencies on $S$ and $H$. The paper achieves this goal by introducing the pessimistic constraint constant $\tau_k' = \tau-\Delta_k$ for the empirical CMDP, to compensate for the underestimation of the cumulative cost by the optimism principle.

**Strengths:**

1. The paper is well-written. The motivation and the proof sketch are clear for the reader.
2. The theoretical contribution is solid. I check the proof of Theorem 4.1 and it is correct at least for me.
3. A detailed comparison (Table 1) is provided, which is great for the reader to get the contribution of this paper.

**Weaknesses:**

1. Even if a detailed comparison of the result is provided, it is unclear in the paper how the author achieves this improvement technically. From my perspective, the improvement arises from the selection of the empirical CMDP. In fact, the definition of the empirical CMDP is different from the previous work, since it uses the optimism principle and compensates the underestimation of the cumulative cost by subtracting the safety threshold $\tau$ by some gap $\Delta_k$. Do I understand correctly? Additional clarification about (a) the motivation for selecting this gap and (b) what's the advantage of this gap technically would enhance the paper.

2. Compared to the previous work with known $\pi^0$ and $c^0$ in Table 1, their algorithms can lead to zero violation with higher regret in theory, so the contribution of this paper mainly focuses on reducing the regret result. Is this true in the experiment? The paper could benefit from containing some empirical results to show the advantage. I understand that for a theory paper, the most important contribution is the theoretical insight, but it is also important to conduct a simple empirical validation of the insight. An empirical simulation and comparison of previous work are enough. I would like to increase my score if the authors provide some empirical results.

3. Many RL problems contain a large or even continuous state space. Can this algorithm be extended to the continuous setting such as linear MDP? If it's not, what are the primary challenges?

**Questions:**

The questions are contained in the weaknesses part.

---

### Official Review · Reviewer_BBFL · 2024-11-03

**Soundness:** 3
**Presentation:** 3
**Contribution:** 3
**Rating:** 5
**Confidence:** 3

**Summary:**

This paper presents a low-switch primal-dual algorithm for safety RL. Tight bounds on regret and safety are achieved.

**Strengths:**

+ A new approach with a low switch with improved bounds on regret and safety.

**Weaknesses:**

-Both approach (i.e., low switching) and analysis are heavily based on the reference Zhang et al. (2024). This casts doubts on how much original novelty comes from this paper.

**Questions:**

what are the major differences between this paper and the reference Zhang et al. (2024), regarding both algorithm design and analysis?

In this paper, it claims ``in large-scale, complex environments. " However, it is hard to argue that a tabular MDP is complex enough.

While this paper focuses on theoretical analysis, it will still help by showing some experiments to validate the theoretical results.

It will be interesting to see which part of the algorithm makes the constrained violation at $\tilde{O}(1)$.

Due to the trade-off between return maximization and constraint satisfaction, it is not straightforward to compare two algorithms. As constrained violation is allowed here, are the lagrangian multipliers zeros here? If not, can we compare the Lagrangian function, as a weighted sum of return and constraint, to see which algorithm works better? Or in general, with constraints unsatisfied, I wonder what are the Karush–Kuhn–Tucker conditions in this case?

---

### Official Review · Reviewer_M5io · 2024-11-05

**Soundness:** 2
**Presentation:** 2
**Contribution:** 1
**Rating:** 5
**Confidence:** 3

**Summary:**

This paper tackles safe reinforcement learning (RL) by introducing a low-switching primal-dual algorithm, termed SLIM (Safe Low-Switching Primal-Dual Model-Based Algorithm). The algorithm addresses a key challenge in reinforcement learning: balancing the trade-off between minimizing regret and ensuring safety within Constrained Markov Decision Processes (CMDPs). Unlike existing methods that frequently switch policies, SLIM aims to achieve safety and performance with minimal policy changes, optimizing computational efficiency and adaptability. Leveraging a Bernstein-based exploration bonus and low-switching model updates, SLIM achieves improved theoretical bounds on both regret and constraint violations, offering a scalable solution for safe RL in complex environments.

**Strengths:**

It seems that the proposed Safe Low-Switching Primal-Dual Model-Based Algorithm achieves a regret bound that reduces the dependency on the state-action space and planning horizon, making it more effective in large-scale environments.

**Weaknesses:**

1. Low-switching appears to be one of the central focuses of the paper; however, I noticed a lack of analysis or formal guarantees supporting the claimed benefits of low-switching. It would be beneficial for the authors to clarify how low-switching is quantified and its impact on the model. For example, could the authors provide a quantitative measure of policy switching frequency in SLIM?

2. The rationale behind how low-switching leads to improved dependency in the regret bound is unclear. A more thorough explanation or intuition on why reduced switching would contribute to lower regret could strengthen the understanding of the proposed approach. For example, could you provide a step-by-step explanation or an illustrative example of how reduced switching translates to improved regret bounds?

3. When comparing the proposed regret results with existing methods, the contribution seems somewhat limited. The differences in assumptions and constraint violation conditions make it challenging to directly assess the relative significance of the improvements, as they may not hold under comparable settings.

**Questions:**

See weaknesses

---

### Meta-Review · Area_Chair_YknP · 2024-12-20

**Metareview:**

The paper proposes a primal-dual algorithm for tabular CMDPs with unknown transitions and deterministic known rewards/costs. The algorithm requires access to a policy that satisfies the constraints (a safe policy). The authors prove a sublinear regret bound (with improved dependency on the number of states and the horizon) and constant constraint violation for the algorithm.

While the reviewers agree on the importance of the problem studied in the paper, i.e., safe decision-making, and like the provided comparison (in terms of regret and constraint violation) with the existing results, they believe the paper requires further improvement, including

(-) The motivation behind several decisions in designing the algorithm is missing.
(-) It'd be helpful if the authors clearly explain the main design decisions that resulted in improvement (in terms of regret bound) in their algorithm (how low switching results in better regret).
(-) Lack of any empirical results to validate the theoretical findings.
(-) Better comparison with the existing algorithms, especially those by Liu et al. (2021) and Bura et al. (2022) that have similar setting (assumptions) as the method proposed in the paper.

**Additional Comments On Reviewer Discussion:**

There was no discussion as the authors did not participate in the rebuttal.

---

### Decision · Program_Chairs · 2025-01-22

Reject